**Trajectory encounter volume as a diagnostic of mixing potential in fluid flows**
Irina I. Rypina[1] and Larry J. Pratt[1]
[1] Woods Hole Oceanographic Institution, Physical Oceanography department,
266 Woods Hole rd., Woods Hole MA, 02543
Corresponding author email: irypina@whoi.edu
Abstract
Fluid parcels can exchange water properties when coming in contact with each other, leading to
mixing. The trajectory encounter mass and a related simplified quantity, encounter volume, are
introduced as a measure of the mixing potential of a flow. Encounter volume quantifies the
volume of fluid that passes close to a reference trajectory over a finite time interval. Regions
characterized by low encounter volume, such as cores of coherent eddies, have low mixing
potential, whereas turbulent or chaotic regions characterized by large encounter volume have
high mixing potential. The encounter volume diagnostic is used to characterize mixing potential
in 3 flows of increasing complexity: the Duffing Oscillator, the Bickley Jet, and the altimetry-
based velocity in the Gulf Stream Extension region. An additional example is presented in which
the encounter volume is combined with the $u^*$ -approach of Pratt et al., 2016 to characterize the
mixing potential for a specific tracer distribution in the Bickley Jet flow. Analytical relationships
are derived connecting encounter volume to shear and strain rates for linear shear and linear
strain flows, respectively. It is shown that in both flows the encounter volume is proportional to
time.
I.    Encounter volume
22         a.   main idea

Mixing is an irreversible exchange of properties between different water masses. This process is
important for maintaining the oceanic large-scale stratification and general circulation, and it
plays a key role in the redistribution of bio-geo-chemical tracers throughout the world oceans.
Mixing occurs between different water masses when they come in direct contact with each other.
Thus, mixing potential of the flow, i.e., the opportunity for mixing to occur, is generally
enhanced in regions where water parcels meet or encounter many other water parcels and thus
are exposed to a large amount of fluid passing by them as the flow evolves. This would be the
case, for example, for a parcel within a chaotic zone –a region of the flow that is in a state of
chaotic advection. There, the separation between initially nearby water parcels grows
exponentially in time and, in the infinite time limit, each water parcel encounters all the other
water parcels within the same zone and gets in contact with the entire volume of the chaotic
zone. Similarly, high encounter volumes will exist in turbulent regions. In contrast, mixing
potential and encounter volume is expected to be smaller in regions where water parcels do not
experience many encounters with other water parcels and remain close to their initial neighbors
as the flow evolves. This would be the case, for example, for a water parcel that is located inside
a coherent eddy. If the eddy is in a state of solid body rotation, the water parcel would forever
stay close to its initial neighbors and will not have any new encounters at all. If some amount of
azimuthal shear is present within the eddy, then for a water parcel located at a radius r from the
eddy center, encounters will be limited to those water parcels located within a circular strip
centered at the same r.
Of course, the presence of a mixing potential does not guarantee that the mixing of a tracer will
occur: it is also essential that the tracer distribution is non-uniform, so that irreversible property
exchange can take place between different water parcels during their encounters. This exchange
happens by diffusion and therefore relies on a concentration difference between two parcels.
Thus, the intensity of mixing would depend on both the tracer distribution and the flow, whereas
mixing potential is the property of only the flow field alone. In this work we introduce the
concept of an *encounter mass, M,* and *encounter volume, V*, which serves as a simplified
representation of $M$ in incompressible flows, as an objective measures of encounters between
different fluid elements in order to quantify the mixing potential of a fluid flow. There are many
existing trajectory-based measures of fluid stirring; ours has the virtue of having a
straightforward physical interpretation and being easy to implement and immediately applicable
to ocean float and drifter data.  Our method does not require sophisticated book keeping as in
braid theory (Allshouse and Thiffeault, 2012) or finite-time entropy (Froyland and Padberg-
Gehle, 2012).

57         b.   definition and numerical implementation

For a given reference trajectory, $\vec{x}(\vec{x}_0, t_0; T)$, the *encounter mass, $M(\vec{x}_0, t_0; T)$*, is defined as the
total mass of fluid that passes within a radius $R$ of reference trajectory over a finite time interval
$to < t < to + T$. One might imagine a sphere that has radius $R$ and that is centered at and moves
with the reference trajectory. The encounter mass then consists of the mass of the fluid that is
initially located within the sphere along with the mass of all the fluid that passes through the
sphere over the time interval $to < t < to + T$. Note that it is generally not possible to compute
the latter by simply integrating the mass flux into the sphere over $to < t < to + T$ since some
fluid may leave and then re-enter the sphere and would be counted more than once, so
Lagrangian information is required to keep track of the history of each fluid parcel trajectory
entering the sphere.
To this end, subdivide the entire fluid at $t = to$ into small compact fluid elements with masses
$\delta M_i = \rho_i \delta V_i$, where $\rho_i$ is the density of a fluid element and $\delta V_i$ is its volume. We wish to follow
the motion of each fluid element over time interval $to < t < to + T$, and we assume that the
elements remain compact over such time, so that the motion of each fluid element can be well-
represented by one trajectory. If the fluid elements stretch and deform too much, we can evoke
the continuum hypothesis and make $\delta M$ sufficiently small that such compactness is assured. In
the limit of infinitesimal fluid elements, $\delta M_i \to dM$, we can associate with each infinitesimal fluid
element a unique trajectory. The encounter mass is then

$$M = \lim_{dM_i \to 0} \Sigma_i \, dM_i.$$

For an incompressible flow, the density and volume of each fluid element, $\rho_i$ and $\delta V_i$, remain
constant following a trajectory, although different fluid elements are still allowed to have
different densities such as, for example, in stratified 3D geophysical flows. If the flow is
unstratified, the densities of all fluid elements are equal, $\rho_i = \rho$, and the encounter mass
becomes

$$M = \rho V,$$

where

$$V(\vec{x}_0, t_0; T) = \lim_{dV_i \to 0} \Sigma_i \, dV_i$$

is the *encounter volume* – the total volume of fluid that passes within a radius $R$ of reference
trajectory over a finite time interval $to < t < to + T$. When all volume elements are equal,
$dV_i = dV$, the encounter volume can be further simplified to

$$V = \lim_{dV \to 0} N dV,$$

where the *encounter number*, $N(\vec{x}_0, t_0; T)$, is the number of trajectories that come within a radius
$R$ of the reference trajectory over a time interval $to < t < to + T$. We will refer to $t_0$ as the
starting time, $T$ as the trajectory integration time, and $\vec{x}_0$ as the trajectory initial position, i.e.,
$\vec{x}(\vec{x}_0, t_0; T = 0) = \vec{x}_0$. For practical applications with geophysical flows, the limit in the
definition of the encounter volume can be dropped and one can simply use the approximation

$$V \approx N \, \delta V$$

with the dense grid of initial positions $\vec{x}_0$. Mathematically, the encounter number can be written
as $N = \sum_{k=1}^{K} I(\min(|\vec{x_k}(\vec{x}_0, t_0; T) - \vec{x}(\vec{x}_0, t_0; T)|) \leq R)$ where the indicator function I is 1 if true
and 0 if false, and K is the total number of Lagrangian particles released. The encounter volume
depends on the starting time, integration time, encounter radius, and the number of trajectories
(i.e., grid spacing); all of these parameter dependences will be discussed below. Once the
encounter volume is estimated, regions of space with large/small $V$ would then be associated
with enhanced/inhibited mixing potential. For the remainder of this paper, we will focus on
incompressible fluid flows and will be concerned with the encounter volume, rather than
encounter mass.

We define $V(\vec{x}_0, t_0; T)$ and $N(\vec{x}_0, t_0; T)$ based on the number of encounters with different
trajectories, not the total number of encounter events, so even if some trajectory first comes close
to the reference trajectory, then moves away and then re-approaches it again later, it is only
counted once.  In a flow field with no sources or sinks of tracer variance, where variance is
therefore decaying, it is reasonable to expect that most property exchange between two parcels
will often occur during their first encounter, thus the motive for counting only the first encounter.
Note that this assumption may not hold if the parcels re-acquire different properties after their
first encounter due to encountering and exchanging properties with other parcels. In this case, or
in the case when tracer variance is being continuously introduced, it may be more reasonable to
count the total number of encounters.
For a numerical implementation of the trajectory encounter volume-based mixing
characterization, one would need to start, at some time $t_0$, with a grid of initial positions
spanning the flow domain, and then evolve trajectories under the flow field over the time interval
$T$. This time interval should be chosen based on the physical properties of the flow and with
specific scientific questions in mind. For example, if the research focus is on ocean submesoscale
dynamics, the time scale $T$ would be on the order of hours to days, whereas the corresponding
time scale for mesoscale dynamics would be on the order of weeks to months.
$V(\vec{x}_0, t_0; T)$ is a Lagrangian quantity that characterizes mixing potential of a flow over a time
interval from $t_0$ to $t_0 + T$. As the flow field evolves in time, its mixing characteristics can
change and $V(\vec{x}_0, t_0; T)$ will reflect this change. For example, if a coherent eddy with weak
mixing potential, embedded in a chaotic zone with enhanced mixing potential, was present in the
flow from time $t_1$ to time $t_2$, but it dispersed and disappeared afterwards, then $V(\vec{x}_0, t_0; T)$ is
expected to be small at those locations $\vec{x}_0$ that correspond to the interior of an eddy for $t_0 \geq t_1$
and $t_0 + T \leq t_2$, whereas for $t_0 > t_2$, when the eddy is no longer present, $V(\vec{x}_0, t_0; T)$ would
increase. Dependences on $T$ and $t_0$ are similarly expected to be present within a chaotic zone.
In the infinite time limit, $T \to \infty$, when all parcels within a chaotic zone (or turbulent region) of
finite extent encounter all other parcels within the same chaotic zone, the encounter volume
$V(\vec{x}_0, t_0; T \to \infty)$ approaches a constant equal to the volume (or area in 2d) of the chaotic zone.
For 2D, incompressible flow, the encounter rates over finite $T$ are locally the largest near a
hyperbolic trajectory and along the segments of its associated stable manifolds. The stable
manifolds serve as pathways that bring water parcels from remote regions into the vicinity of the
hyperbolic trajectory, where parcels stay for extended periods of time, and where many
encounters occur. Note that the unstable manifolds, on the other hand, will rapidly remove a
particle from a hyperbolic region, thus limiting its exposure to the high-encounter region near the
hyperbolic trajectory. For this reason, the unstable manifolds are not revealed by encounter
volume calculation performed in forward time and require a backward-time calculation instead.
This exclusive link between forward/backward in time calculation of trajectories and
stable/unstable manifolds, respectively, is not specific to the encounter volume diagnostic, but
rather is typical for many finite-time methods from the dynamical systems theory, including
finite-time Lyapunov exponents (FTLEs), which in forward time approximate segments of stable
manifold as maximizing ridges (Haller, 2002; Shadden et al., 2005; Lekien and Ross, 2010).
Since locations of hyperbolic trajectories and manifolds generally evolve in time, $V(\vec{x}_0, t_0; T)$ is
expected to also vary with $t_0$. As the trajectory integration time $T$ increases, water parcels
initially located further from the hyperbolic trajectory will have the opportunity to come into its
vicinity along the stable manifold. Such parcels, as they approach the hyperbolic trajectory, are
expected to have more encounters than their neighbors that are initially located off the manifold
and thus bypass the vicinity of the hyperbolic trajectory where many encounters occur. Thus,
$V(\vec{x}_0, t_0; T)$ reveals longer segments of stable manifolds for longer integration time $T$, as will be
illustrated numerically in the next section. In the long integration time limit, when each
manifold, either stable or unstable, densely fills the entire chaotic zone forming a dense
homoclininc or heteroclinic tangle, the whole tangle will be characterized by high encounter
volumes in both forward and backward time. Again, this is similar to how the maximizing ridges
of the forward time FTLEs elongate and sharpen with increasing integration time.
The radius $R$, which defines how close to a reference trajectory should another trajectory come in
order to be counted as an encounter, is an important parameter for the calculation of the
encounter volume V. Generally, $R$ should be small compared to the spatial scale of the smallest
features of interest. Specifically, for the $V$ field to delineate a flow feature, say, an eddy,
trajectories within the eddy interior should not encounter those on its exterior. The boundary
region near the eddy perimeter, where such encounters can occur, has the width $2R$. So, if that
width is comparable to or larger than the eddy size, then the eddy would get completely smeared
out and will not be resolved. From a practical viewpoint, however, using very small $R$ would
require very dense grids of trajectories to be computed, otherwise zero or very small number of
trajectory encounters will occur in the entire flow domain. Numerical examples in the next
section suggest that choosing $R$ to be a fraction, up to about half of the size of the smallest
features of interest work best.
Finally, the approximation $V \approx N \, \delta V$ breaks down for sparse grids of initial positions with the
insufficient number of Lagrangian particles, when $N$ is small and $\delta V$ is large. It also works
poorly when applied to 2D divergent flows due to $\delta V$ changing following trajectories. Numerical
simulations in the next section suggest that grid spacing $\leq R/2$ is sufficient, and that the method
can also be applied to characterize mixing potential in slightly divergent two-dimensional flows.
Once the time scale $T$ is identified, grid of initial positions is chosen, trajectories are computed,
radius $R$ is defined, and the number of encounters, $N(\vec{x}_0, t_0; t)$, is counted for each trajectory,
then the encounter volume can be estimated as $V \approx N \, \delta V$ and plotted as a function of the
trajectory initial position $\vec{x}_0$. The resulting $V$ field delineates the flow regions with different
mixing properties as subdomains having different values of $V$.

## II.    Examples

We proceed to test the performance of the encounter volume technique in quantifying mixing potential for several geophysically relevant sample flows of increasing complexity, starting from a simple analytically prescribed periodically perturbed double-gyre Duffing Oscillator system, followed by a dynamically consistent solution of the PV conservation equation on a beta-plane known as the Bickley Jet, and finishing with an observationally based geostrophic velocity field in the North Atlantic derived from the sea surface height altimetry.

### a.    Duffing Oscillator

The Duffing Oscillator flow and its figure-eight geometry has become a standard test case for emerging techniques related to the dynamical systems theory. This flow consists of two gyres with the same sign of rotation (clockwise in our case), whose elliptic centers oscillate in time around their mean position. A hyperbolic point is located at the origin between the two gyres, and a pair of stable and unstable manifolds emanate from it forming a figure eight in the absence of the time dependent perturbation, or forming a classic homoclinic tangle in the presence of the perturbation. The velocity field is two-dimensional and incompressible and is given by $u = y$ and $v = (x - ax^3)(1 + \epsilon \cos(\omega t + \phi))$ with $a = 1$, $\omega = 3\pi/2$, $\phi = \pi/4$ and $\epsilon = 0.1$. With these parameters, the Poincare section (Fig. 1 bottom) shows the presence of two main regular elliptic regions with O(1) radius corresponding to the interiors of the gyres, which are embedded into a figure-eight shaped chaotic zone, within which a number of island chains with smaller regular islands are present. The winding time for most trajectories in the system is on the order of $5T_{pert}$ with $T_{pert} = \frac{2\pi}{\omega}$, except for trajectories near the hyperbolic point for which winding time is much longer (Fig. 1 top).

The encounter volume was computed for a range of trajectory integration times, from $T = T_{pert}$ (which is significantly shorter than trajectory winding time) to $T = 50T_{pert}$ (significantly longer than trajectory winding time), and for a range of encounter radii, from $R = 0.01 \ll R_{eddy}$ (significantly smaller than the eddy core radius) to $R = 1 \approx R_{eddy}$ (comparable to the eddy core radius). The results in Fig. 2 suggest that the encounter volume method works best for integration times longer than the trajectory winding time and encounter radius about 1/3 to 1/2 of the gyre radius (right 3 panels of the middle row). For very small encounter radius (top row), V is noisy because trajectories simply do not encounter many neighbors. Thus, delineating the domain into regions with different mixing potential, as in the top right panel, requires long integration time. For $T = 50T_{pert}$, good agreement with the Poincare section is observed, and the use of small encounter radius allows for a precise identification of smaller regular island chains, such as the chains of 4 islands located just outside of the perimeter of both left and right eddy cores. Note that the noise in the V field can be suppressed by using a denser initial grid of trajectories, but at the cost of a more expensive computation. For very short integration times (left column) when trajectory segments are very short, the encounter volume is not capturing the

difference between the regular and chaotic regions. This is not surprising as velocity shear is probably a dominating factor over such small times. As the integration time increases, the difference in encounter volume becomes more pronounced between trajectories that remain within the eddy cores and trajectories that are free to move around the chaotic zone. Over a time scale of approximately one winding period (or about 5 periods of the perturbation; second column), the two regular eddy cores (blue regions with small $V$) and a segment of the stable manifold (red curve emanating from the origin with largest $V$) becomes clearly visible for R=0.2 and R=1. The revealed manifold segment becomes longer, narrower and more tangled, eventually filling up the whole chaotic zone. At the same time, the shape of the core region becomes more exact and approaches the "true" core in the Poincare section as the integration time increases to 50 periods of the perturbation. The agreement with Poincare section is excellent in the right middle panel, although the smaller island chains are not as visible as in the top right panel because of the use of a larger encounter radius that is comparable to their size (see Fig. 3). Finally, for the large encounter radius that is comparable to the size of the eddy (bottom row), the boundary region near perimeter of an eddy, within which trajectories on the inside of the eddy can encounter trajectories passing by on the outside, is as wide as the eddy itself, essentially wiping out all small scales from the V field. All of these trends are in agreement with theoretical expectations described in Section I.

In order to more clearly highlight the link between high values of $V$ and stable (rather than unstable) manifolds, we have computed both stable and unstable manifolds for the Duffing Oscillator flow using a direct method, where we grew manifolds from a small segment starting at the hyperbolic trajectory. For the Duffing Oscillator this computation is straightforward since the the hyperbolic trajectory stays at the origin at all times. Both stable and unstable directly-computed manifolds were then superimposed on a forward-time encounter volume plot in Fig. 4. The comparison shows that, as anticipated, the encounter volume diagnostic clearly highlights stable manifolds as maximizing ridges of $V$ computed in forward time.

With a variety of dynamical systems techniques available, it is important to understand the advantages and limitation of the different methods. We compared the encounter volume to two well-established and commonly-used methods, the Poincare section (Fig. 3) and the FTLEs (Fig. 5). Since the Poincare section requires stroboscopic sampling of trajectories in time, it can only be applied to time-periodic flows, and requires that trajectories are computed over long integration time, typically thousands of the periods of the perturbation. On the other hand, it generally requires only a few parcels to be released at some key locations, rather than releasing a dense grid of initial positions, to map out the entire phase space. The encounter volume and FTLEs, on the other hand, are not limited to time-periodic flows, and also work with significantly shorter segments of trajectories (longest integration time in our simulations in Fig. 2 is only 50 periods of perturbation). They are also better suited for identifying manifolds than the Poincare sectioning as they do not require any *a priori* knowledge about the location of the

hyperbolic trajectory. On the other hand, they require many more parcels to be released in order
to map out the phase space. When applied to the same set of trajectories (same initial positions
and integration times), the FTLEs and the encounter volume methods produced similar results
(Fig. 5), with $V$ being arguably better suited for 1) identifying the coherent core regions of
eddies, where FTLEs have spiraling patterns that complicate the analysis, and 2) producing more
continuous segments of manifolds at intermediate integration times, when FTLE-based ridges get
discontinuous near the turning points of a manifold. The advantage of FTLEs, on the other hand,
is that they have fewer parameters ($T$ and grid spacing), whereas $V$ also depends on $R$, and that
they less expensive computationally. The more expensive computational cost of $V$ compared to
FTLEs is due to two reasons: first, the FTLEs only depend on the initial and final positions of
trajectories, whereas $V$ depends on the entire trajectory history; and second, FTLEs depend on
the relative distance between a trajectory and its closest neighbors, whereas $V$ keeps tracks of
encounters with all trajectories, not just the neighboring trajectories. Thus, the cost of evaluating
FTLE for each particle is independent of the total number of particles released, and the cost of
evaluating $V$ for each particle increases in proportion to the number of particles (since one needs
to keep track of encounters with all particles). The calculation of $V$ is still feasible for realistic
geophysical flows, as is illustrated below. Note also that, depending on the physical question
being studied, the information about the entire trajectory, not just the final and initial position,
might in fact be advantageous.
Related to issue of computational cost is the question of a sufficient grid size. We have carried
out numerical simulations (Fig. 6) to investigate the dependence of the encounter volume on the
grid size, and to come up with a rule of thumb recommendation regarding the appropriate grid
spacing. Our simulations suggest that the encounter volume values (approximated by $V \approx$
$N\,dV$) are relatively insensitive to the variations of grid spacing between 1/10 and 1/2 of the
encounter radius (with the encounter radius being a fraction of the size of the feature of interest,
as suggested by Fig. 2), and that the major effect of a coarser grid is the degraded resolution of
the resulting V map, rather than incorrect V values.

286         b.  Bickley Jet

The meandering Bickley jet flow is an idealized, but linearly dynamically consistent, model for
the eastward zonal jet in the Earth's Stratosphere (del-Castillo-Negrete and Morrison, 1993;
Rypina et al., 2007a;  Rypina et al., 2011). This flow consists of a steady eastward zonal jet on
which two eastward propagating Rossby-like waves are superimposed. All flow parameters used
here are identical to those used in our previous 2007 and 2011 papers. In the reference frame
moving at a speed of one of the waves, the flow consists of a steady background velocity subject
to a time periodic perturbation. The background looks like a meandering jet, with three
recirculation gyres to the north and south of the jet core. Between the recirculation gyres, there
are three hyperbolic points with the associated stable and unstable manifolds. Under the
influence of the time-periodic perturbation imposed by the second wave, heteroclinic tangles are
formed by the manifolds emanating from different hyperbolic regions between the recirculations,
and a chaotic zone emerges on either side of the jet. The manifolds, however, cannot penetrate
through the jet core, which remains regular and acts as a transport barrier separating the northern
and southern chaotic zones. All of these features are clearly visible in the Poincare section shown
in Fig. 4 (top). The bottom subplot shows the V field computed using the encounter radius
$R=5*10^5$, which is about half of the recirculation region radius, and using trajectory integration
time on the order of a few winding times within the recirculations. As expected, the encounter
volume identified 6 recirculation regions and the jet core as zones with small mixing potential
(blue). 6 blue recirculation regions are embedded into two distinct chaotic zones with enhanced
mixing potential (yellow-red) on either side of the jet. Mixing potential is the largest (red) along
the segments of stable manifolds emanating from the hyperbolic trajectories between
recirculations.

          309                 c.   Altimetry-based velocity in the meandering Gulf Stream region

Past its separation point from the coast at Cape Hatteras, the strong and narrow Gulf Stream
current turns off-shore, where it loses its coherence, broadens and weakens, and starts to
meander. Some of the meanders then grow and eventually detach from the current forming
strong mesoscale eddies known as the Gulf Stream rings. On 11 July, 1997 a number of such
Gulf Stream rings of various strength and size and at different stages of their lifetime were
clearly present both north and south of the Gulf Stream Extension Current (Fig. 7).
The flow in the Gulf Stream Extension region, with a non-steady meandering jet and the Gulf
Stream rings and recirculations to the north and south of the jet core, has a lot in common, at
least qualitatively, with the Bickley Jet example. Unlike the idealized model, however, the real
Gulf Stream rings have finite lifetimes, and the jet is not periodic in the zonal direction.
Nevertheless, many of the qualitative features of the Bickley Jet's $V$ field hold in this example.
Specifically, trajectories inside coherent eddy cores have smaller encounter volumes than the
eddy peripheries, and the jet centerline has smaller encounter volume than the flanks.
The velocity field that we used was downloaded from the AVISO website
(http://www.aviso.altimetry.fr/en/data/products/sea-surface-height-products/global.html) and
corresponds to their gridded product with ¼ deg spatial resolution and temporal step of 1 day.
This velocity is based on the altimetric sea surface height measurements made from satellites.
The heights were converted into velocities using geostrophic approximation. For the encounter
volume estimation, trajectories were seeded on a regular grid with $dx = dy \cong 0.06$ deg on 11
July 1997 and were integrated forward in time for 90 days using a fifth-order variable-step
Runge-Kutta integration scheme with bi-linear interpolation between grid points in space and
time. The encounter radius was chosen to be 0.3 deg, which is about a third of the radius of a
typical 200-meter-wide Gulf Stream ring.
The encounter volume was estimated for three different integration times, $T=$ 30 days, 60 days
and 90 days (Fig. 7). The $V$ field clearly indicates that a number of Gulf Stream rings were
present on both sides of the meandering jet. Among those, two strongest ones can be seen at
54W, 36N and 52W, 41N, with the low-$V$ (blue) core and high-$V$ (red) periphery. As the
integration time increases from 30 days to 90 days, the Gulf Stream rings generally start to leak
fluid, their cores start to lose coherence, and the encounter volume within eddy cores starts to
increase as more and more trajectories escape into the eddy surroundings over time. After a 90
day integration time, only a few Gulf Stream rings still possess coherent cores, whereas others
become leaky throughout. Even for the two strongest rings, the coherent Lagrangian cores
(bluish regions with $V \approx 0$) shrink down in size and, importantly, become significantly smaller
than what the Eulerian velocity field would suggest. The core of the northern eddy also gets
shifted slightly to the east from the corresponding Eulerian stagnation point, and becomes
deformed into a non-convex sickle-like shape.
The overall leakiness of the Gulf Stream rings and the small extent of their coherent Lagrangian
core regions suggests that the coherent transport by the Gulf stream rings (and maybe by
mesoscale eddies in general) over time intervals of a few months or longer may be significantly
smaller than what is generally anticipated from Eulerian diagnostics based on closed streamlines
or Okubo-Weiss type criteria. Interestingly, the prominent red rings (large $V$ values) around the
eddy cores in Fig. 7 indicate that significant contribution to transport by Lagrangian eddies may
be due to the high-mixing-potential peripheries rather than the coherent cores themselves.
To visualize the Lagrangian evolution of the core regions and to illustrate the eddy leakiness, we
extracted trajectories from the core of the northern eddy in Fig. 7(left) (i.e., trajectories with
$V < 6000$ km$^2$ from the 30-day-long $V$ field), and plotted their subsequent positions after 30
days, 60 days and 90 days. The results in Fig. 8 confirm that the eddy core stays completely
coherent over 30 days (i.e., all trajectories stay together), but starts to deteriorate at 60 days, with
only a small fraction of the initial patch still staying together and the rest of the patch dispersing
and forming long and narrow filaments.
The jet region, although noisy, seems to suggest higher $V$ near the flanks and smaller $V$ near the
centerline. The center region is not as well-defined as in the Bickley Jet example, possibly
because the Gulf Stream inhibits but does not fully prevent the meridional transport in this
region, and because our encounter radius might have been too large to reveal the central region,
if the true center region was narrower than $2R$ (0.6 degrees). Finally, the $V$ field suggests that the
mixing potential of the flow is not symmetric with respect to the jet centerline and is higher on
the northern side. It would be interesting to see if this is a general property of the flow in this
region or if this phenomenon is specific to the time interval chosen. This investigation is left for
future study.

III.    Encounter volume for some simple flow regimes
By analogy with molecular diffusion, eddy diffusivity, $K$, is often used to characterize the eddy-
induced downgradient tracer transfer in realistic geophysical fluid flows (LaCasce 2008; Vallis,
2006; Rypina et al., 2015; Kamenkovich et al., 2016). Because of the simplicity of this approach,
the majority of existing non-eddy-resolving oceanic numerical models are diffusion based,
despite the somewhat questionable assumptions underlying this approach. An analytical
connection between the encounter volume and diffusivity would thus be useful for the
parameterizations in numerical models.
Although we have not been able to find an analytical expression connecting $V$ and $K$, we outline
below some steps in that direction that help framing the problem. Let us start by considering a
simple diffusive random walk particle motion in two-dimensions, where particles take steps of
fixed length $L$ in random directions at time intervals $\Delta t$. For such process, the single particle
dispersion,
$D = < (x - x_0)^2 + (y - y_0)^2 >,$
which characterizes the mean square displacement from the particle's initial position $(x_0, y_0)$,
grows in proportion to the number of steps, $n$, i.e.,
$D = Kn\Delta t,$
with the proportionality coefficient, $K = L^2 / \Delta t,$ denoting the diffusivity. The angular brackets
denote ensemble average. We are interested in finding an analytical expression for the encounter
number, i.e., the number of particles that pass within radius $R$ from a reference particle over time
$T$, as a function of $K$ and $T$.
It is convenient to move to a reference frame that is tied to a reference particle, which would then
always stay at the origin, while other particles would be involved in a random walk motion. The
problem of finding the encounter number then reduces to counting the number of particles that
come within radius $R$ from the origin over time $T$ in the moving frame. The properties of the
random walk process in the moving reference frame are different from those in the stationary
frame. Specifically, the direction of each step in the moving reference frame still remains random
(since it is a sum of two random variables, each uniformly distributed within an interval $[0; 2\pi]$),
but the step size is no longer fixed. Instead, the step size can be written as
$L_m^2 = dx_m^2 + dy_m^2 = (dx - dx_{ref})^2 + (dy - dy_{ref})^2 = 2L^2 - 2(dx\, dx_{ref} + dy\, dy_{ref}),$
where $dx$ and $dy$ correspond to displacements of a particle in x and y directions at some instance
in time, and subscripts $m$ and $ref$ denote the moving reference frame and the reference trajectory,
respectively. Denoting the angle in which the step is taken by $\varphi$, the displacements are $dx =$
$L\cos\varphi, dy = L\sin\varphi, dx_{ref} = L\cos\varphi_{ref}, dy_{ref} = L\sin\varphi_{ref}$ leading to
$L_m = 2L\sin\alpha$, where $\alpha = \frac{\varphi-\varphi_{ref}}{2}$. Since both $\varphi$ and $\varphi_{ref}$ are random variables uniformly
distributed between 0 and $2\pi$, $\alpha$ is a random variable with a flat pdf distribution $\in [0;\pi]$.
This change in the step size between the stationary and moving frames leads to a doubling of the
diffusivity in the moving reference frame. To show this, we write down the dispersion in the
moving frame as

$$D_m =< \left(x_m - x_{0_m}\right)^2 + \left(y_m - y_{0_m}\right)^2 >=$$

$$=< \left(x - x_{ref} - x_0 - x_{0ref}\right)^2 + \left(y - y_{ref} - y_0 - y_{0ref}\right)^2 > =$$

$$= D - 2\,\Delta x_{ref} < \Delta x > -2\Delta y_{ref} < \Delta y > +\Delta x_{ref}^2 + \Delta y_{ref}^2 =$$

$$= D + \Delta x_{ref}^2 + \Delta y_{ref}^2,$$

where $\Delta x = x - x_0$ is the deviation from the initial position in the stationary frame and similarly
for $\Delta y, \Delta x_{ref}$ and $\Delta y_{ref}$. We have used $< \Delta x >=< \Delta y >= 0$ to get the last equality. When
averaged over many reference trajectories, $<\Delta x_{ref}^2 + \Delta y_{ref}^2 >= D$ since in the stationary
reference frame the reference particle is doing a random walk just like all other particles, so that
$< D_m >= 2D$, or, equivalently, $< K_m >= 2K$.
We thus seek an expression for the number of particles that are involved in a random walk
process with diffusivity $2K$ and that come within an encounter radius $R$ from the origin during
their first $n$ steps ($n$ plays the role of discretized integration time). This quantity is related to the
first passage time density, which characterizes the probability that a particle has first reached an
absorbing boundary (often referred to as a cliff in statistics) at time $t$, and its integral quantity,
called the survival probability, which characterizes the probability that a particle has not come in
contact with absorbing boundary over time $t$ (i.e., it survived after time $t$ without falling off a
cliff). So far, however, we have not been able to complete the derivation and we leave this
development for a future investigation.
Numerical Monte-Carlo simulations of a random walk process suggest that the dependence of
the encounter number (and encounter volume) on the integration time $T$ is not a linear and not a
square-root function. The power-low least square fit of the form $V\sim T^\alpha$ returns $\alpha$ values between
0.64 and 0.78 for a wide variety of $R$ and $K$, each spanning an order of magnitude interval of
values. Similarly, the power-low least square fit $V\sim K^\beta$ and $V\sim R^\gamma$ yield $\beta \cong 0.664$ and $\gamma \cong$
431 0.69.

The ballistic spreading that is dominated by a local velocity shear is another commonly-
encountered spreading regime. There, the separation between particles grows in proportion to
time. Ballistic spreading can often be observed in nonsteady realistic oceanic flows at time scales
that are much shorter than the onset of diffusive spreading (which develops after a trajectory
samples multiple different eddies or other flow features). To derive a connection between
encounter volume and velocity shear, consider a trajectory that is advected by a flow field with
constant meridional velocity shear, $\gamma$, of zonal velocity. In a reference frame moving with a
reference trajectory the velocity profile is, $u(y) = \gamma y$ where u denotes the x-component of
velocity, and the encounter volume becomes
$$V \cong Ndxdy = 2\int_0^R dy \int_R^{R+x(T)} dx = 2\int_0^R dy \int_0^T u(y)dt = \gamma R^2 T, \tag{2}$$
suggesting a linear growth with time for a ballistic regime. Note that expression (2) quantifies the
encounter volume as a volume of fluid that is initially located outside of the encounter sphere
and that passes through the sphere over time $T$. To include the volume of fluid that is initially
located within the encounter sphere (or within the encounter circle in this 2D case), one needs to
add $\pi R^2$ to expression (2). The contribution of this term gets negligibly small as $T \rightarrow \infty$.
Expression (2) has been tested numerically and shows good agreement with the numerically-
estimated encounter volume for a linear shear flow (Fig. 10(right)).
The steady linear saddle flow with a constant strain rate $\alpha$ and velocities
$$u = \alpha x; v = -\alpha y. \tag{3}$$
is another commonly-considered example often used to approximate the vicinity of a hyperbolic
trajectory in more complicated non-steady non-linear situations. A unique property of this flow is
that the velocity profile is unchanged in any reference frame moving with a trajectory. This can
be shown by applying the coordinate transformation, $\hat{x} = x - x_{tr}(t); \hat{y} = y - y_{tr}(t)$, where
$(x; y)$ are coordinates in a stationary frame, $(\hat{x}; \hat{y})$ are coordinates in a moving frame, and
$(x_{tr}(t); y_{tr}(t))$ is the trajectory. The velocity in a moving frame is then
$$\hat{u} = u - \frac{dx_{tr}}{dt} = \alpha x - \frac{dx_{tr}}{dt} = \alpha\hat{x} + \alpha x_{tr} - \frac{dx_{tr}}{dt} = \alpha\hat{x}$$
$$\hat{v} = v - \frac{dy_{tr}}{dt} = -\alpha y - \frac{dy_{tr}}{dt} = -\alpha\hat{y} - \alpha y_{tr} - \frac{dy_{tr}}{dt} = -\alpha\hat{y} \tag{4}$$

where the last equality holds because $\frac{dx_{tr}}{dt} = \alpha x_{tr}; \frac{dy_{tr}}{dt} = -\alpha y_{tr}$. Thus, without loss of
generality, we can consider a flow in a reference frame moving with a reference trajectory that is
located at the origin. The encounter volume that comes within a radius R of the origin over the
time interval $T$ can be written as
$V \cong Ndxdy = \int_0^T F_\perp(t)dt,$

463      (5)

where $dx$ and $dy$ denote the grid spacing between neighboring trajectories, and the flux of
trajectories entering the circle is given by
$F_\perp = \int u_\perp ds.$      (6)
Again, as in our treatment of the linear shear flow, expression (5) does not include the volume of
fluid that is initially located within the encounter sphere (or encounter circle in this 2D case), but
only the volume that was initially located outside but passes through the sphere over time $T$. The
contribution of that fixed volume ($\pi R^2$), gets negligibly small as $T->\infty$. Here $u_\perp$ is the inward-
looking normal component of velocity at a circle of radius R, and $ds$ is an infinitesimal segment
of the circle arc. From symmetry, the flux is the same in each of the 4 quadrants so we will
consider the 1$^{st}$ quadrant only. From geometry (Fig. 11),
$u_\perp = -u\sin\beta - v\cos\beta = \alpha R(\cos^2\beta - \sin^2\beta)$ and $ds = Rd\beta$, leading to
$F_\perp{}^{1st\ quad} = \alpha R^2 \int_0^{\pi/4}(\cos^2\beta - \sin^2\beta)\ d\beta = \frac{\alpha R^2}{2}$      (7)
and
$V^{1st\ quad} = \int_0^T F_\perp(t)dt = \alpha R^2 T/2.$      (8)
Adding the other 3 quadrants then gives
$V = 2\alpha R^2 T.$      (9)
Numerical simulations of the encounter volume in a linear strain flow show excellent agreement
with theoretical expression (9) (Fig. 10(left)).
The linear growth of the encounter volume with time in the linear shear and linear strain flows
could be anticipated by noting that both flows are steady in a reference frame moving with a
reference trajectory, and all particles only encounter the origin once and never come back. Thus,
the flux through the encounter circle is constant in time and the encounter volume, which is a
time-integral of flux, is proportional to time. The random walk flow seems to be different
because the particles can encounter the reference trajectory more than once, leading to a non-
steady flux of first encounters and a non- linear time dependence of the encounter volume.
IV.     Mixing potential for a specified tracer: the $\boldsymbol{u}^*$-approach
The above examples are centered on mixing potential of a flow field, but there may be value in
computing the encounter volume for swarms of trajectories of biological organisms, drifting
sensors, and other non-Lagrangian trajectories.  For example, if one is interested in the actual
transport of scalar properties such as heat, salt, or vorticity, then it may be useful to calculate V
using a velocity field that is directly linked to the vector flux of the scalar of interest. This
approach has been used in connection with heat transport in advective/diffusive flow (Bejan,
1995; Costa, 2006; Mahmud and Fraser 2007; Mukhopadhyay et al., 2002, and Speetjens, 2012)
and more recently with the transport of more general scalars in forced and dissipative (and
possibly turbulent) flows (Pratt et al., 2016). The central idea is to a define velocity field $\boldsymbol{u}^*$
based on the (known) flux $\boldsymbol{F}$ of a scalar with concentration $C$. Here bold quantities denote
vectors. The concentration is assumed to obey a conservation equation of the form
$$\frac{\partial C}{\partial t} = -\nabla \cdot \boldsymbol{F} + S, \tag{10}$$
where S contains the sources and sinks of $C$.  The velocity $\boldsymbol{u}^*$ is defined as the velocity of a
hypothetical flow in which the flux of $C$ is purely advective: $\boldsymbol{F} = C\boldsymbol{u}^*$. Pratt et al., 2016 show
that, in the absence of sources or sinks of $C$, that the total amount of $C$ contained within any
material boundary advected by this hypothetical flow is conserved: $\frac{d}{dt}\int_V C dV = 0$. Thus $\boldsymbol{u}^*$ is
linked to scalar property fluxes while $\boldsymbol{u}$ is limited to fluid volume (or area) fluxes.
If indeed $\boldsymbol{F}$ is due entirely to advection by the actual fluid velocity field $\boldsymbol{u}$, then $\boldsymbol{u}^*=\boldsymbol{u}$, but more
generally $\boldsymbol{F}$ will contain contributions from eddy fluxes, molecular or sub-grid diffusion, and
even forcing and dissipation terms that can be expressed as the divergence of a flux. In addition,
$\boldsymbol{F}$ may be augmented by the addition of any non-divergent vector without altering Eq. (3).  As
shown by Speetjens (2012), this lack of uniqueness can be dealt with by defining a physically
relevant reference scalar distribution and then focusing on the flux of the scalar anomaly, an
approach we adapt below. Thus, by estimating the encounter volume V for trajectories of the $\boldsymbol{u}^*$
field, one is quantifying the rate at which different 'parcels' of tracer anomaly are brought into
contact with each other. An example is presented next.

516        a.   Example: encounter volume for a tracer with a specified initial distribution in a
517             Bickley Jet flow

In this subsection we apply the encounter volume diagnostic to quantify the mixing potential for
a specific tracer in the Bickley Jet flow. Our goal is to describe an example where the $\boldsymbol{u}^*$ field for
a given tracer is significantly different from the flow velocity $\boldsymbol{u}$, and where the corresponding
encounter volume field for a given tracer, $V^*$, is significantly different from the water particle
trajectory-based encounter volume V.
Consider the Bickley Jet flow with the same parameters as in II(b) and assume that one is
interested in a tracer that, at initial time t0, has uniform *value* $c_0$  south of the jet and has a
constant meridional gradient north of the jet, i.e., $C_0 = c_0 + 0.5y(sign(y - 5*10^5) + 1)$ with
$c_0 = 1$. Ignoring the diffusive terms, the tracer evolution is governed by the advection
equation $\frac{\partial C}{\partial t} = -\nabla(\boldsymbol{u} \cdot C)$, where $\boldsymbol{u}$ is the Bickley Jet flow velocity. Since the jet core acts as a
transport barrier separating the northern and southern chaotic zones, this tracer will rapidly
filament and develop high property gradients north of the jet, but will remain uniform south of
the jet. So, despite the fact that the mixing potential of the Bickley Jet flow is exactly the same
on both sides of the jet (Fig. 7(bottom)), stirring will not lead to mixing for this particular tracer
distribution south of the jet, where tracer gradient is zero, thus leading to zero mixing potential
for this particular tracer. We seek to capture this effect via applying the encounter volume-based
mixing diagnostic to the corresponding $\boldsymbol{u}*$ field for this tracer.
In the spirit of Speetjens (2016) we regard $c_0$ as the reference concentration, here constant, and
define $\boldsymbol{F}$ to be the flux of a tracer anomaly: $\boldsymbol{F} = \boldsymbol{u} \cdot (C - c_0)$. The resulting $\boldsymbol{u}^* = \frac{\boldsymbol{F}}{C} = \boldsymbol{u}\left(1 - \frac{c_0}{C}\right)$
is zero south of the jet where $C = c_0$ and is approximately equal to $\boldsymbol{u}$ north of the jet where
$C \gg c_0$, leading to the $\boldsymbol{u}*$ -based encounter number $V^* = 0$ south of the jet and $V^* \approx V$ north of
the jet.
This behavior was further validated numerically in Fig. 12, where we first numerically simulated
the evolution of this tracer in the Bickley Jet flow, then estimated $\boldsymbol{u}*$, counted $N^*$ and estimated
$V \cong NdV$ for trajectories advected by the $\boldsymbol{u}*$ field. The result confirms that mixing potential for
this tracer is zero south of the jet, $V^* = 0$, whereas north of the jet $V^*$ is very close to $V$ from
Fig. 7(bottom). Thus, by combining the $\boldsymbol{u}*$ approach with the encounter volume idea, we were
able to correctly capture the mixing potential for a specific tracer.
V.      Summary and discussion
When water parcels come in direct contact with each other, they can exchange water properties,
leading to mixing. The trajectory encounter volume, $V$, quantifies the volume of fluid that passes
close to a reference trajectory over a time interval $t_0 < t < t_0 + T$. Thus, the encounter volume
is proportional to, and can be used as a measure of, the mixing potential of a flow. For
incompressible flows densely seeded with particles, the encounter volume can be approximated
by $V \cong N\delta V$ where $N$ is the encounter number, i.e., the number of trajectories that come come
within radius $R$ from the reference trajectory over time $t_0 < t < t_0 + T$, and $\delta V$ is a small
volume element.
The encounter volume diagnostic was tested in 3 flows with increasing complexity, the Duffing
Oscillator, the Bickley Jet, and the altimetry-based velocity in the Gulf Stream Extension region.
In all cases, $V$ was smaller within cores of coherent eddies and jets, where mixing potential was
low, and $V$ was larger in chaotic zones near the peripheries of the eddies and at the flanks of the
meandering jets, where the mixing potential of the flow was high.
Similar to finite-time Lyapunov exponents (FTLEs) that are commonly used to delineate regions
with qualitatively different motion (Haller, 2002; Shadden et al., 2005; Lekien and Ross, 2010),
$V$ depends on the trajectory starting time, $t_0$, allowing tracking the evolution of oceanic features
by repeating the calculation at different $t_0$, and on the trajectory integration time, $T$, revealing
different structures that impact the mixing potential of the flow from time $t_0$ to time $t_0 + T$.
Specifically, longer segments of stable/unstable manifolds emanating from hyperbolic regions
are revealed for longer $T$ in forward/backward time. In the long-$T$ limit, when both the stable and
unstable manifolds densely fill the entire chaotic zone, $V$ approaches a constant equaling to the
volume of the chaotic zone.
$V$ also depends on the encounter radius $R$, which defines how close two trajectories need to be in
order to be counted as an encounter. Analytic arguments and numerical simulations both suggest
that $R$ on the order of a fraction (~1/3) of the radius of the smallest feature of interest should
work well in most cases.
Finally, while $V$ was initially introduced in the continuous limit of infinitely many infinitely
small fluid elements (i.e., infinitely dense grid of initial positions), its approximation $V \cong N\delta t$
depends on the initial spacing between neighboring trajectories. Numerical simulations suggest
that this approximation works well for grid spacing as large as $R/2$ (with the appropriately
chosen $R$ as discussed above), and that the major effect of increasing the grid spacing is in the
degraded resolution of the resulting $V$-map rather than incorrect $V$ values.
As with FTLEs, complexity measures (Rypina et al., 2011), Lagrangian descriptors (Mendoza et
al., 2014) and other techniques from the dynamical systems theory (Beron-Vera et al., 2013;
Budisic and Mezic, 2012; Froyland et al., 2007; Haller et al., 2016), $V$ can be computed for
forward and backward in time trajectories, with the backward computation revealing unstable
manifolds. Our encounter number could plausibly be related, in a limiting case, to the mixing
geometry of Karrash and Keller, 2017.
For a ballistic spreading regime dominated by the velocity shear $\gamma$, and for the linear saddle flow
with a constant strain $\alpha$, $V$ was shown to be proportional to $\gamma t$ and $\alpha t$, respectively. The linear
growth of the encounter number with time for the linear shear and linear strain flows is a
consequence of the steady flux of first encounters through the encounter circle.
An analytical connection between the encounter volume and a widely-used measure of mixing,
the diffusivity $K$, would be a desirable result for parameterizing the effects of eddies in
numerical models. Some initial developments towards deriving such a formula were outlined for
a diffusive random walk process. It was shown numerically that the dependence of $V$ on time is
non-linear, but numerical simulations were too inconclusive to make further inferences.
The mixing potential is the property of the flow field and characterizes the intensity of stirring,
whereas the actual tracer mixing depends both on the flow and the tracer. For example, no tracer
mixing will occur if the tracer gradient is zero, even if the mixing potential of the flow is high.
To address this, we have proposed combining the encounter number diagnostic with the $\boldsymbol{u}^*$-
approach of Pratt et al, 2016 for characterizing the mixing potential for a specific tracer $C$. $\boldsymbol{u}*$
depends on, and includes information about, the tracer fluxes. In the absence of sources and sinks
of $C$, the amount of tracer is conserved within any Lagrangian volume advected by $\boldsymbol{u}*$, so the
encounter volume $V^*$ computed for trajectories advected by $\boldsymbol{u}*$ can be used to quantify the
mixing potential for a specific tracer. An example was presented where $V^*$ for a specified tracer
distribution in the Bickley Jet flow was significantly different from $V$ in a large part of the
domain.
The encounter volume is a frame-independent quantity because it is based on relative distances
between water parcel trajectories, rather than on properties of isolated trajectories. The encounter
volume values do not change under orthogonal transformations of coordinates, i.e., under
rotations and translations of a reference frame. This is a desirable property because the ability of
a flow to mix tracers should not depend on the reference frame.
The encounter volume and, more generally, encounter mass ideas presented in this paper are not
restricted to two dimensions and can be used to quantify mixing potential in three-dimensional
flows. This framework also does not require incompressibility and can work with unstructured
irregular grids. The investigation of the performance of the method in quantifying mixing
potential of a flow in such more complicated cases is left for a future study.
**Acknowledgments:** This work was supported by the NSF grants OCE-1154641, OCE-1558806
and EAR-1520825, ONR grant N00014-11-10087 and NASA grant NNX14AH29G.

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

**Figure 1. Trajectory segments for different integration times (top) and Poincare section**
**(bottom) for the Duffing Oscillator**

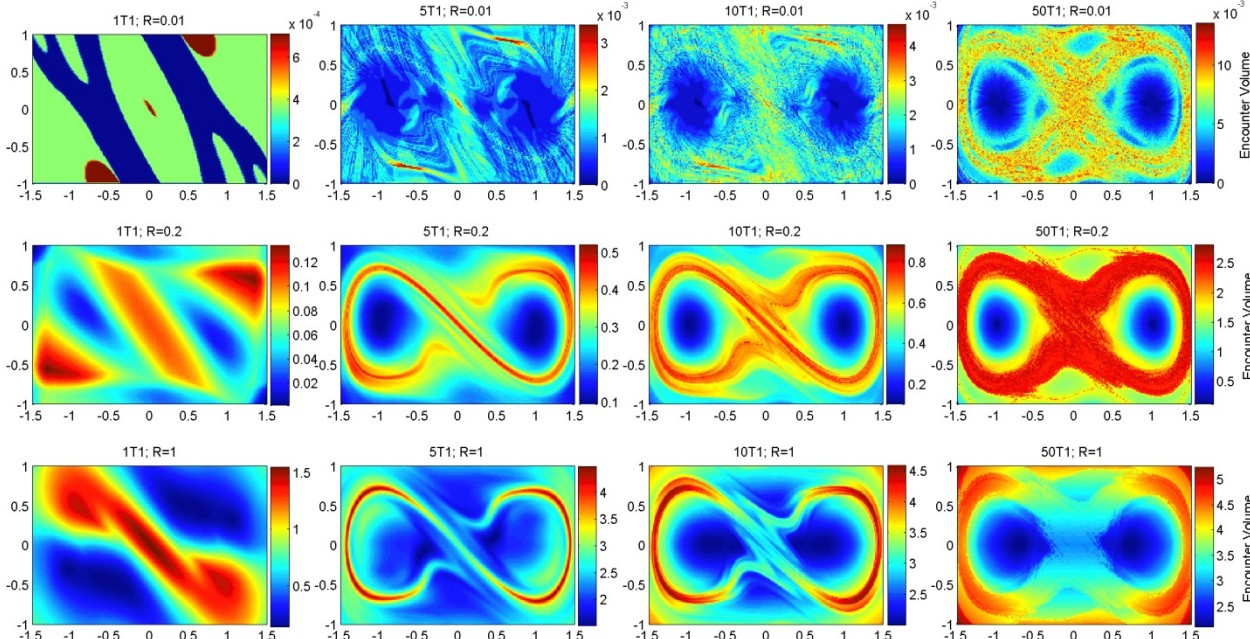

**Figure 2. Encounter volume for the Duffing Oscillator for various integration times,**
**from T=0.1Tpert (on the left) to T=50Tpert (on the right), and for various encounter**
**radii, from R=0.01 (on the top) to R=1 (on the bottom). Trajectories were released on a**
**regular grid spanning the entire domain with grid spacing of 0.013 in both x and y**
**directions.**

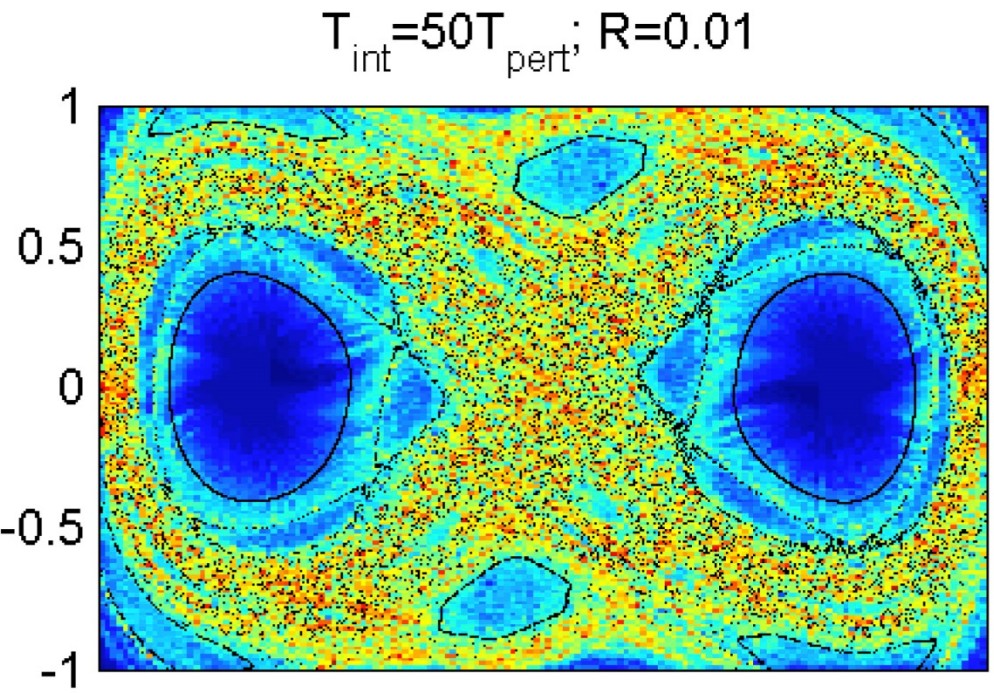

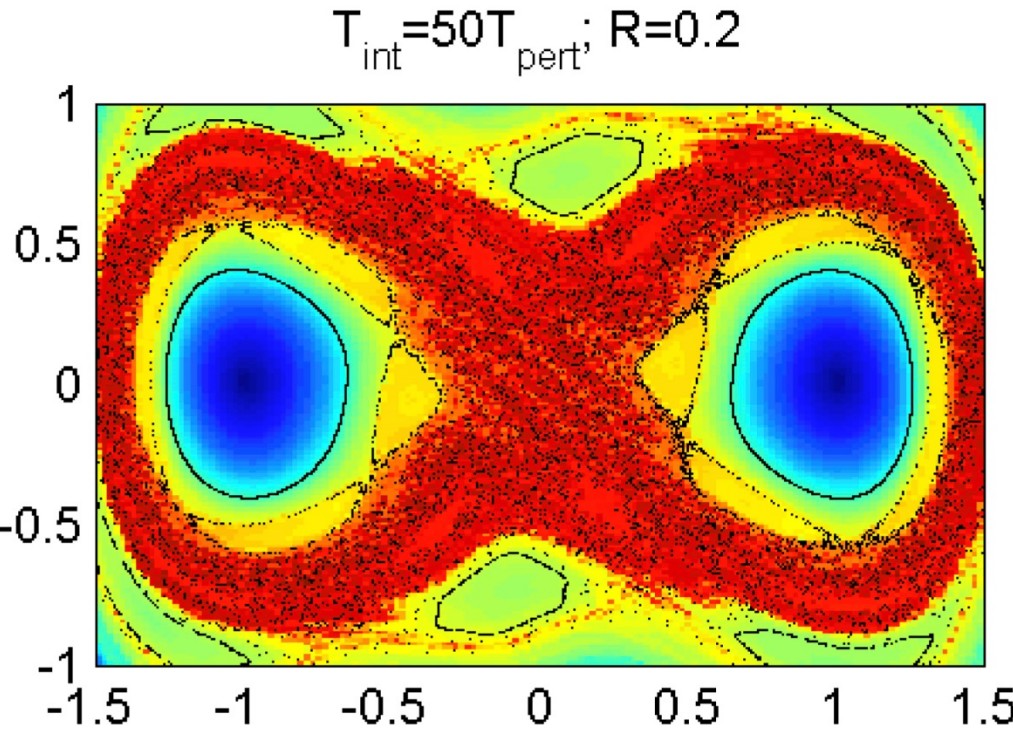

**Figure 3. Poincare section (black dots; same as in the bottom panel of Fig. 1) superimposed onto the encounter volume (color; same as top and middle right panels in Fig. 2). Only select trajectories from the Poincare section are shown.**

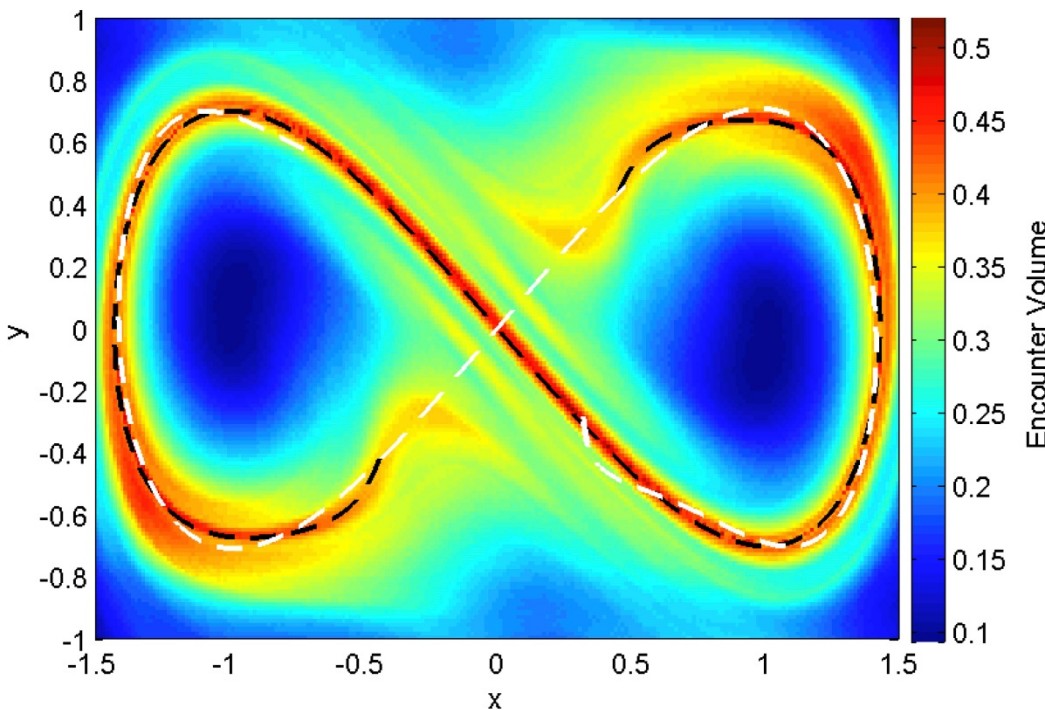

**Figure 4. Encounter Volume (color; the same as 2[nd] row and 2[nd] column subplot of Fig. 2)**
**and stable (black) and unstable (white) manifolds for the Duffing Oscillator flow computed**
**using the direct method.**












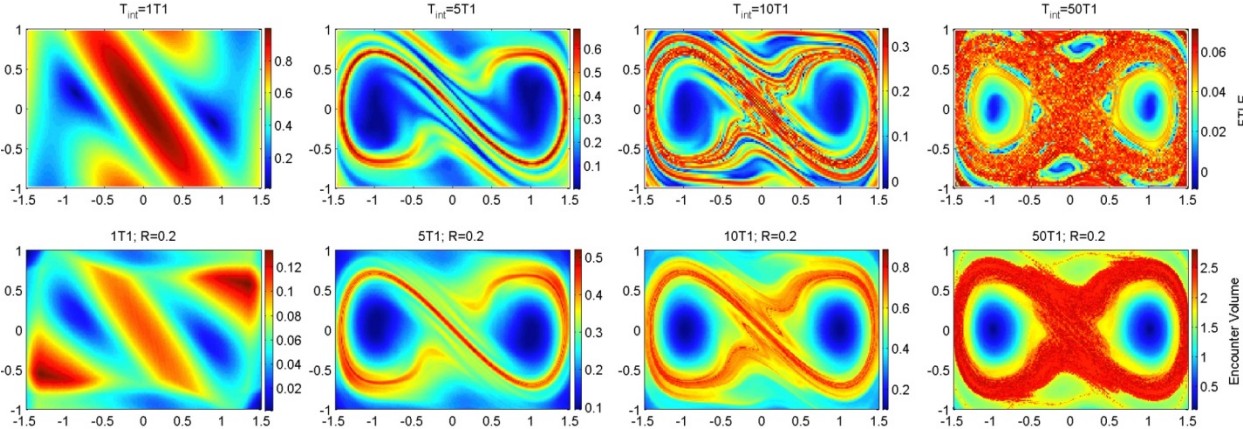

Figure 5. Comparison between the FTLEs (top) and the encounter volume (bottom; same as middle row of Fig. 2) for the Duffing Oscillator flow for various integration times, from T=0.1Tpert= 0.13 (on the left) to T=50Tpert=66.67 (on the right). The same set of trajectories, deployed on a dense initial grid with 0.02 grid spacing is used in all simulations. In the bottom panels, R=0.2.

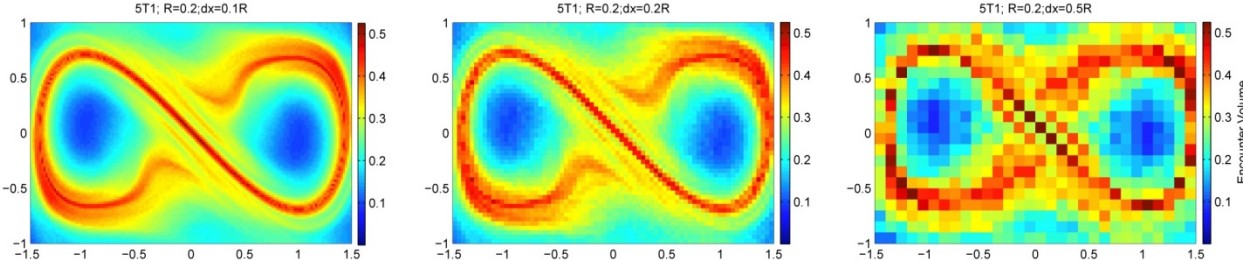

**Figure 6. Encounter volume, V, for the Duffing Oscillator flow for various grids of initial**
**positions, from dense grid spacing of 0.02 (left), to intermediate grid spacing of 0.04**
**(middle), to coarse grid spacing of 0.1 (right). Encounter radius, R=0.2, and integration**
**time, T=6.67, are the same in all 3 simulations.**



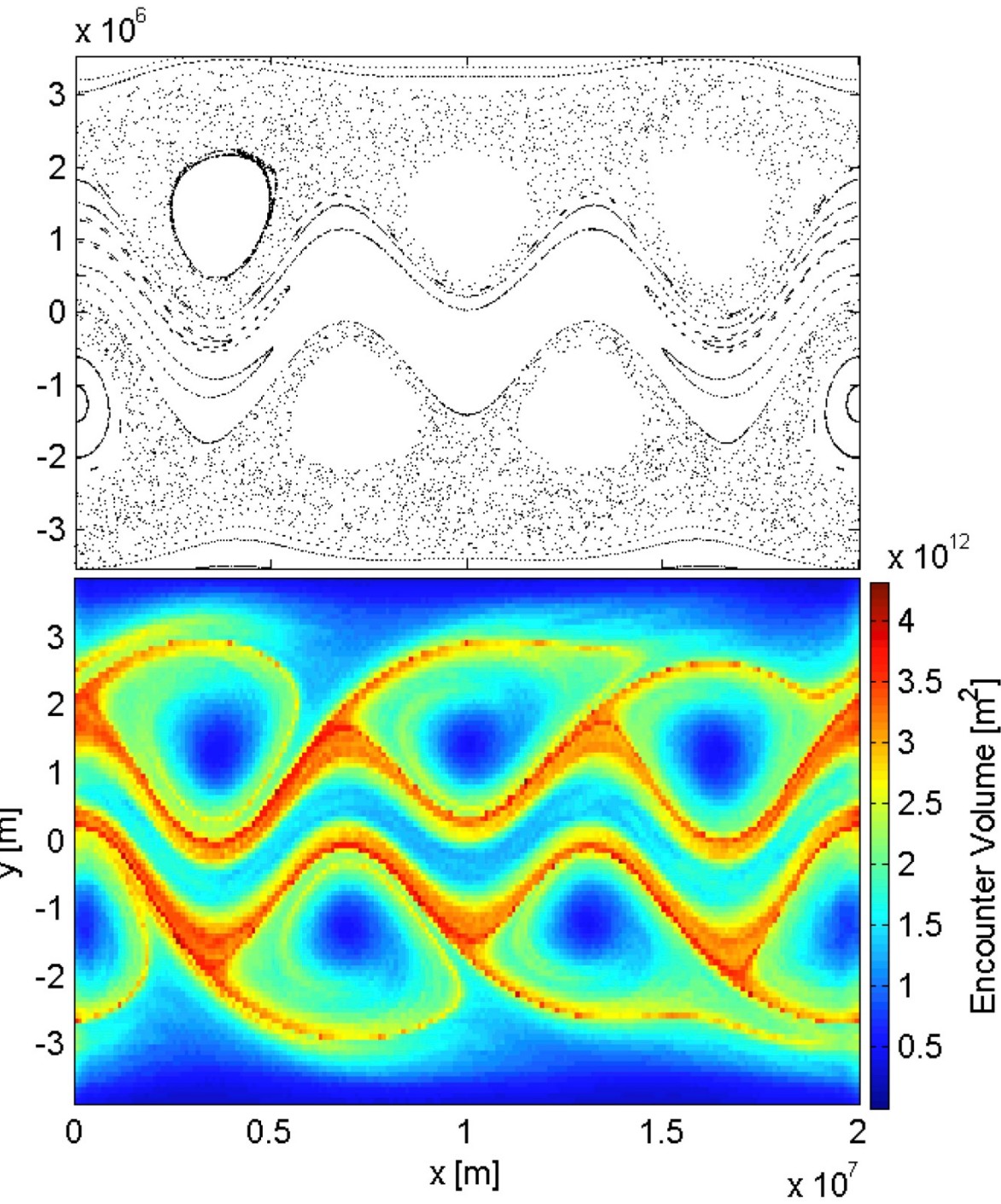

**Figure 7. Poincare section (top) and encounter volume V (bottom) for the Bickley Jet flow. For the V calculation, trajectories were released on a regular grid spanning the entire domain with grid spacing of about $10^5$ in both x and y directions.**

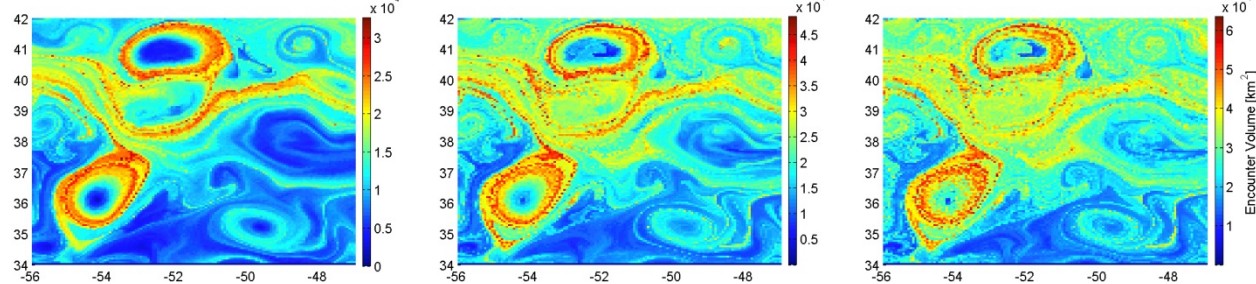

**Figure 8. Encounter volume for the AVISO velocities in the Gulf Stream Extension region for trajectories released on 7/11/1997 and integrated over 30 days (left), 60 days (middle) and 90 days (right). Trajectories were released on a regular grid spanning the domain from 65W to 35W and from 30N to 50N with grid spacing of about 0.06 deg in both longitude and latitude. Additional simulations were performed to insure that the release domain was sufficiently large, and that further increase of the release domain does not lead to changes in the encounter volume for trajectories starting in the subdomain shown.**

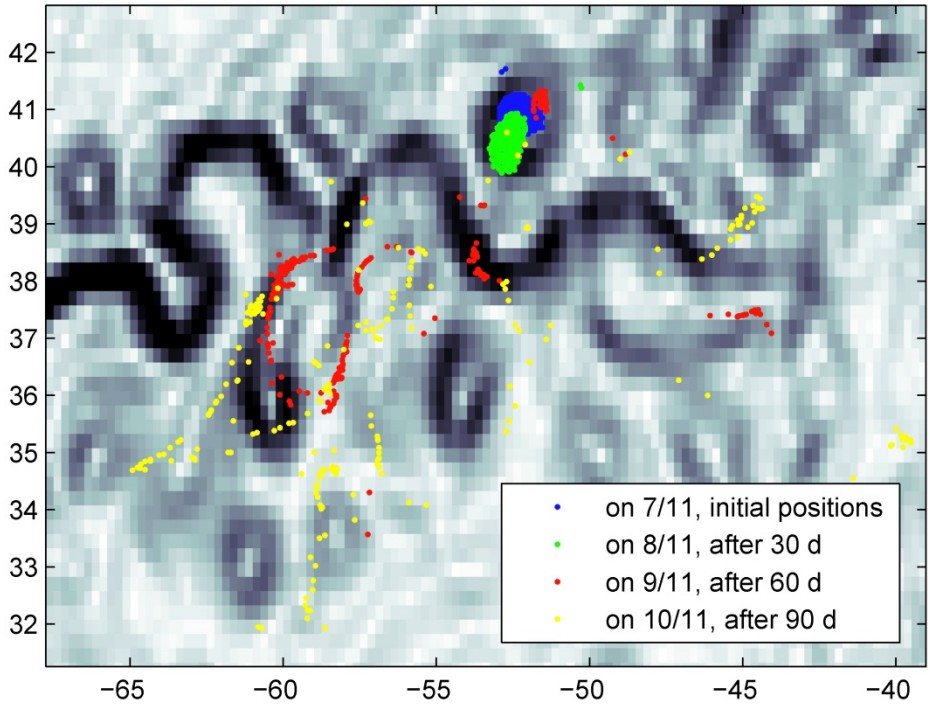

**Figure 9. Positions of trajectories that were initially located within the eddy core on 7/11/1997 (blue patch) after 30 days (green), 60 days (red) and 90 days (yellow) of integration. Background shows the flow kinetic energy snapshot on 7/11/1997.**

















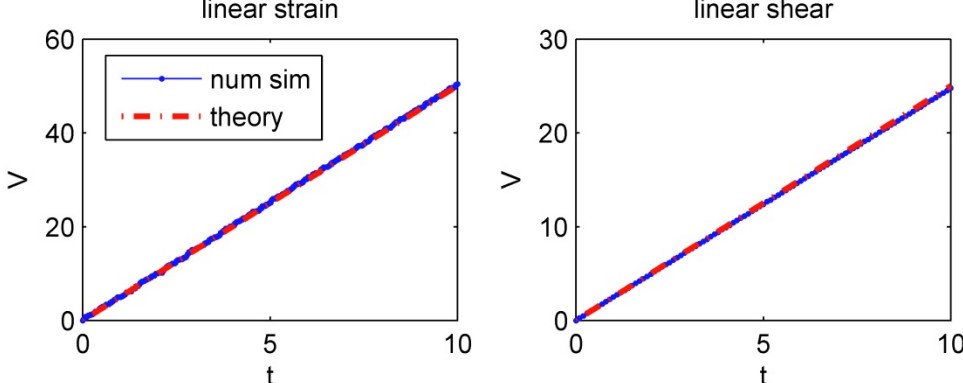

**Figure 10. Comparison between numerically computed encounter volume (blue) and analytical predictions (eqs. (8) and (9)) (red) for the linear strain (left) and linear shear flows (right). For the linear shear flow alpha=0.1, R=5, dx=dy=R/25; for the linear strain flow gamma=0.1, R=5; dx=dy=R/25. Other parameter choices show good agreement as well.**

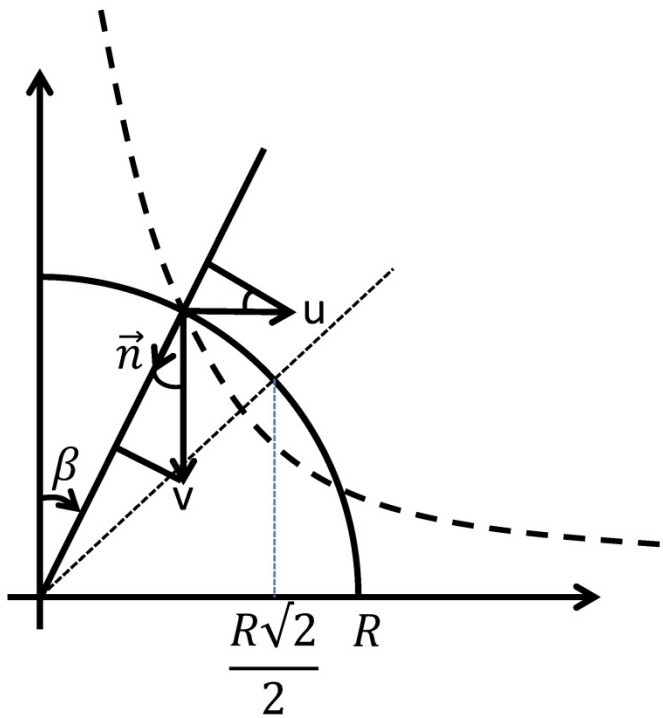


**Figure 11. Schematic diagram for estimating encounter number for a linear saddle.**




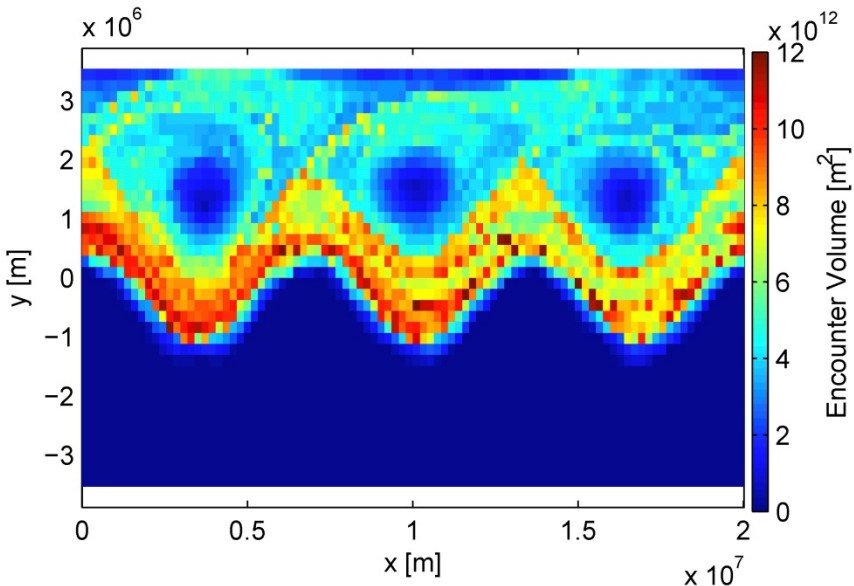


**Figure 12.** $u*$-**based encounter volume,** $V^*$, **for a tracer with un initial distribution south**
**the jet and constant meridional gradient north the jet.**


