# Peer review of "Trajectory encounter volume as a diagnostic of mixing potential in fluid flows"

_Nonlinear Processes in Geophysics, 2016_

## Referee Comment (RC1) · Anonymous Referee #1 · 2 Jan 2017

**Manuscript**  NPG-2016-72

**Title**  Trajectory encounter number as a diagnostic of mixing potential in fluid flows

**Authors**  Irina I. Rypina and Larry J. Pratt

**General comments**  The study proposes a new diagnostic for evaluation of the mixing potential of fluid flows: the trajectory encounter number. This diagnostic is for a given trajectory defined as the number of other trajectories it approaches to within a pre-defined distance during a specific time interval. The new diagnostic is demonstrated by way of two analytical flows and a data-based flow. The proposed approach is certainly of interest for mixing analyses and, due to its straightforward concept and structure, seems particularly suited for data-based studies. Moreover, the manuscript overall is well written. However, a number of scientific and technical issues arise that must be addressed in a revision in order for the manuscript to become acceptable for publication. Details are below.

**Specific comments**

1. Line 43: "... property exchange can take place between different water parcels ..." Mention that this exchange happens by diffusion and therefore relies on a concentration difference between two parcels. The relevance of tracer non-uniformity and the fact that mixing potential alone may not suffice is then evident.

2. Lines 49–50: "Our method does not require the initial spacing between trajectories to be small ..." Mustn't the spacing always be sufficiently small to detect the relevant spatial features that determine the mixing properties? In other words, doesn't your method therefore require comparable spacings as other methods in order to properly capture the physics? Your examples in fact employ fairly dense spacings (see also remark below). Please comment.

3. Lines 62–67: encounter number $N$ is determined by the *first* encounter of a given trajectory with other trajectories. This relies on the assumption that in the absence of sources/sinks "most property exchange will occur during the first encounter" and in other cases "... the number of first encounters ... should still be relevant." This is a rather loose argumentation. May the concentration difference between parcel A and B (also in the absence of sources/sinks) not just as well be *larger* – causing *more* property exchange – on e.g. a *second* encounter due to property exchange of parcels and A and B with other parcels in between their first and second encounters? Please provide a stronger physical rationale for this first-encounter ansatz or present it more explicitly as an assumption or hypothesis.

4. Lines 84–86: "... encounter rates ... are locally the largest near a hyperbolic trajectory and along the segments of its associated stable manifolds ..." This exclusive link with stable manifolds is unclear. Don't the high encounter rates result from the rapid dispersion of fluid parcels due to exponential stretching in the homoclinic/heteroclinic tangles delineated by interacting (un)stable manifolds of hyperbolic points? In other words, don't stable and unstable manifolds contribute equally to the high encounter rates in chaotic regions? Hence, it seems more accurate to correlate regions of high $N$ with such tangles instead of only with stable manifolds. Please either better explain the (assumed) role of stable manifolds or link the behaviour with chaotic tangles.

5. Lines 93–94: "... will reveal longer segments of stable manifolds ... illustrated numerically in the next section." It actually more and more seems to reveal the abovementioned homo-clinic/heteroclinic tangles instead of the stable manifolds. Consider to this end the Duffing oscillator in Sec. IIa. Here the stable and unstable manifolds of the hyperbolic point form a pair

symmetric about $x = 0$ (as remarked on line 126). Their interaction yields a homoclinic tangle that delineates a figure-8 region about the two islands in Fig. 1. This tangle – and thereby *both* manifolds – coincides with the region of highest encounter rates in Fig. 2. Results on the Bickley jet in Fig. 4 further seem to support this; here correlation actually occurs with the heteroclinic tangles delineated by the interacting (un)stable manifolds of the 3 hyperbolic points instead of only with the stable manifolds. Please comment and, if necessary, modify the discussion.

6. The discussion of Fig. 2 implies that the encounter number indeed adequately captures the dynamics. However, to this end rather smooth distributions (as e.g. in Figs. 2–4) seem neces­sary, suggesting that the method requires a dense spacing of initial parcel positions in order to work properly. This contradicts the statement "... does not require the initial spacing between trajectories to be small ..." (lines 49-50). Moreover, this suggests that mixing analyses by the encounter number may in fact be far more expensive than standard Poincaré sectioning (typically requiring only a few dozen parcels). Please comment and, if necessary, modify the discussion.

7. The above suggests that Poincaré sectioning outperforms the encounter-number method in pe­riodic flows. Hence, the periodic examples mainly serve to demonstrate the physical validity of the encounter-number method; its true usefulness seems to be for essentially aperiodic flows as e.g. the Gulf stream flow (Sec. IIc). However, the analysis of this flow is rather superficial and open-ended (lines 209–230). It is recommended to deepen this analysis so as to convincingly demonstrate the potential of the method (in particular) for aperiodic flows.

8. Sec. III: it is recommended to demonstrate validity of expressions (1), (2) and (9) for $N$ by comparison with $N$ found via actual parcel trajectories of the corresponding simplified flows.

9. Lines 310–311: "... vector flux of the scalar of interest. This linkage is made explicit by ..." This same concept of a net scalar flux (and corresponding trajectories) is in fact also adopted in studies on convective heat transfer and chemically-reacting flows [1, 2, 3, 4, 5]. Please mention this for a stronger connection with similar research and literature.

10. Line 324: "Although this lack of uniqueness may seem troublesome ..." This ambiguity is in fact resolved in [5] by attaching physical validity to such an additional vector instead of treating it as an arbitrary field (see also remark below).

11. Lines 347–348: "... it is most convenient to make use of the flexibility in the definition of the tracer flux ..." This suggests that the method produces arbitrary results and its physical meaning therefore is questionable. However, this approach can in fact be provided with a sound physical basis using the approach following [5]. Key to this is that, given linear transport equations, a scalar field $c$ governed by a transport equation of the form (10) admits expression as the difference between two other physically-meaningful scalar fields $c_A$ and $c_B$, each governed by

$$\frac{\partial c_A}{\partial t} + \boldsymbol{\nabla} \cdot \boldsymbol{F}_A = S_A, \qquad \frac{\partial c_B}{\partial t} + \boldsymbol{\nabla} \cdot \boldsymbol{F}_B = S_B, \tag{1}$$

with $\boldsymbol{F}_{A,B}$ and $S_{A,B}$ the corresponding fluxes and source terms, respectively. In [5], $S_A = S_B = 0$ and $\boldsymbol{F}_A$ and $\boldsymbol{F}_B$ are diffusive flux and advective-diffusive flux, respectively, of the same initial condition $c_A(\boldsymbol{x}, 0) = c_B(\boldsymbol{x}, 0) = g(\boldsymbol{x})$. In the current manuscript, also $S_A = S_B = 0$ yet $\boldsymbol{F}_A$ and $\boldsymbol{F}_A$ now both are the advective flux (i.e. $\boldsymbol{F}_{A,B} = \boldsymbol{u}c_{A,B}$) of the *different* initial conditions $c_A(\boldsymbol{x}, 0) = c_0$ and $c_B(\boldsymbol{x}, 0) = C_0(\boldsymbol{x})$. Hence, both problems, though physically different, allow for treatment by the same concept. Transport of difference $c' = c_B - c_A$ is governed by

$$\frac{\partial c'}{\partial t} + \boldsymbol{\nabla} \cdot \boldsymbol{F}' = S', \quad \boldsymbol{F}' = \boldsymbol{F}_B - \boldsymbol{F}_A, \quad S' = S_B - S_A, \tag{2}$$

with here $S' = 0$ and $\boldsymbol{F}' = \boldsymbol{u}c' = \boldsymbol{u}(c_B - c_A)$ the flux of $c'$ (i.e. the anomaly and its flux in line 348). Thus anomaly $c'$ in fact concerns the scalar transport relative to a physical reference state $c_A$ instead of some arbitrary state. Here the reference state happens to remain uniform in time due to the advective transport of a uniform initial condition, i.e. $c_A(\boldsymbol{x}, t) = c_A(\boldsymbol{x}, 0) = c_0$, yet the approach holds equally for any non-uniform (evolving) state $c_A$ (enabling its employment also for more complicated problems).[1] Moreover, note that $\boldsymbol{u}$ needn't be divergence-free. It is recommended to modify the discussion in Sec. IV according to the above in order to eliminate the (incorrect) impression of a conceptual flaw in the method.

**Minor technical issues and corrections**

1. Figs. 2–4: specify the spacing of the initial parcel positions.

2. Line 153: pronounces → pronounced

3. Line 231: the title of Sec. III is rather long and confusing. Please consider a more compact title.

4. Line 266: reference moving → reference frame moving

[1] A. Bejan, Convection Heat Transfer, Wiley, New York (1995).

[2] V.A.F. Costa, Bejan's heatlines and masslines for convection visualization and analysis, Appl. Mech. Rev. 59 (2006), 127.

[3] S. Mahmud, R. A. Fraser, Visualizing energy flows through energy streamlines and pathlines, Int. J. Heat Mass Transfer 50 (2007), 3990.

[4] A. Mukhopadhyay, X. Qin, S. K. Aggarwal, I. K. Puri, On extension of "heatline" and "massline" concepts to reacting flows through use of conserved scalars, ASME J. Heat Transfer 124 (2002), 791.

[5] M. F. M. Speetjens, A generalised Lagrangian formalism for thermal analysis of laminar convective heat transfer, Int. J. Therm. Sci. 61 (2012), 79.
* * *
[1]Reference state $c_A$ in [5] e.g. corresponds with the non-uniform and unsteady temperature field due to diffusive heat transfer only; $c' = c_B - c_A$ is the contribution to the total advective-diffusive temperature field $c_B$ due to the flow $\boldsymbol{u}$ (i.e. $c' \neq 0$ only if $\boldsymbol{u} \neq \boldsymbol{0}$) and thus captures the thermal transport that is effectively induced by the fluid motion.

---

## Referee Comment (RC2) · Anonymous Referee #2 · 2 Jan 2017

In this paper the authors introduce a new Lagrangian descriptor to give a measure of the effectiveness of a flow to mix over a finite time. The idea is to start with a finite grid of K initial trajectories, and, for each trajectory compute the number other trajectories that come within a radius R of the given one, thus they compute

$$N(x, R, T, K) = \sum_{k=1}^{K} I(\min_{s,t \in [0,T]}(\|\phi_s(x_k) - \phi_t(x)\|) \le R)$$

where we define the indicator function $I$ to return 1 if true and 0 if false, and the flow $\phi_t(y) = x(t; y)$ for an initial point y. (The authors never give such a formula, and ignore the dependence on the gird).

While this is an intriguing idea, it is not clear how to make it mathematically well-

defined. It seems to have some relation to finite time entropy, as introduced in the reference by Froyland –this computes the growth rate of number of distinguishable trajectories. Would it be better to talk about a growth-rate here too? I feel that one should not just compute something that is so specific to choices, but first make a consistent mathematical definition: something that exists in the limit as the grid of initial points becomes infinitely fine, say, and then compute it, showing that the computations are, to some approximation, giving the desired quantity.

1. The authors fix the grid size and do not investigate how the number depends on grid size. The do not even tell the reader what grids are used in the first two examples!

2. It seems like it would be better to define something that (like they mention in the conclusions) represents a "fraction" or "density" of encounters. Mathematically one would probably define something that uses an $\epsilon - \delta$ construction: Given trajectories on an $\epsilon$ grid, how many get closer than $\delta$? Then take limits. if possible, of a density or growth-rate?

3. Another possible quantity, though instead of measuring "mixing" would be one that measures "ergodicity": How many grid cells does a given trajectory cover? This might also be an interesting quantity, and much easier to compute. Note that mixing is equivalent to each trajectory visiting every grid cell.

4. The authors do not really compare their results with any of the other many possible descriptors like FTLE, or perhaps more relevantly the finite time entropy.

5. The authors do not discuss the complexity of this computation. It seems to me that it is much more computationally intensive than, e.g the FTLE, which does not involve comparing all distances between all trajectories. It this really a feasible calculation? How does it scale with the number of trajectories and the time?

6. The authors do some basic investigation of how $N$ depends upon $R$ and $t$, but the computations of $N$ for the simple diffusive and shear cases seem wrong to me:

In particular, if we take a planar diffusive process with diffusion coefficient $D$, and make the assumption (not clear to me) that one can transform to a frame moving with one particle (doesn't this double the diffusivity?), then one should compute the probability of a particle finding itself inside a disk of radius R for any time $0 < t < T$, given it starts at some point $(x_0, y_0)$ in the plane. For example for the process on the line, then at a FIXED time $t$ this means evaluating the integral

$$P(|x(t)| < R | x(0) = x_0) = \frac{1}{2\sqrt{Dt}} \int_{-R}^{R} \exp\left(-\frac{(x - x_0)^2}{2Dt}\right) dx$$

ï£ijwhich can be evaluated in terms of error functions. The authors seem to assume a deterministic motion with the root mean square distance, which seems to me to be wrong. They also ignore particles that start inside the circle of radius $R$ (not so important if they want a large $t$ limit I suppose). Now to compute $N$ you have to sum (or integrate?) this probability over an initial distribution of initial points, say $x_0$ is uniform on a box, perhaps? And you have to somehow compute the probability over all times $0 < t < T$. This calculation seems very different from the one given in the paper.

The shear flow is easier, but I think not done correctly either. One has to compute the area of the region that sweeps into the circle of radius R, but also include the particles that start inside the disk.

While this paper has an intriguing idea, I think it needs substantial revision and correction before publication.

---

## Short Comment (SC1) · 19 Jan 2017

It is only the short comment, not deep review. We can see many finite time Lagrangian descriptors suggested last time, maybe 20 or 30 years. For example, the Poincare section is calculated for finite time. Most descriptors follow from dynamical systems theory constructions. It well discussed in reviewers comments. I have an analogous question. Is it possible to make some connection between the encounter number and the Poincare recurrence.

In any case, I think, the most Lagrangian descriptors used, for example, in oceanography have no strong mathematical foundation, but very useful for data analyze and interpretation. I also think the descriptor suggested here is more expensive in comparison with FLTE or some other. But it seems, it has some advantage in physical

interpretation.

I think the manuscript is suitable for publication in Nonlinear Processes in Geophysics, after some revision.

I can find few misprints in the manuscript which not mentioned by reviewers.

Minor technical issues and correction

lines 336,371-373

t0 is used and may be better to use $t_0$

Caption to figure 8

U-star and Nstar are used. Maybe it will be better to use $U^*$ and $N^*$

---

## Author Comment (AC1) · 16 Mar 2017

**Manuscript** NPG-2016-72

**Title** Trajectory encounter number as a diagnostic of mixing potential in fluid flows

**Authors** Irina I. Rypina and Larry J. Pratt

**Anonymous Referee #1**

**General comments** The study proposes a new diagnostic for evaluation of the mixing potential of fluid flows: the trajectory encounter number. This diagnostic is for a given trajectory defined as the number of other trajectories it approaches to within a pre-defined distance during a specific time interval. The new diagnostic is demonstrated by way of two analytical flows and a data-based flow. The proposed approach is certainly of interest for mixing analyses and, due to its straightforward concept and structure, seems particularly suited for data-based studies. Moreover, the manuscript overall is well written. However, a number of scientific and technical issues arise that must be addressed in a revision in order for the manuscript to become acceptable for publication. Details are below.

**Specific comments**
1. Line 43: "... property exchange can take place between different water parcels ..." Mention that this exchange happens by diffusion and therefore relies on a concentration difference between two parcels. The relevance of tracer non-uniformity and the fact that mixing potential alone may not suffice is then evident.

*We have included this clarification in the revision. The last paragraph of Sec I(a) now reads: "...*the presence of a mixing potential does not guarantee that the mixing of a tracer will occur: it is also essential that the tracer distribution is non-uniform, so that irreversible property exchange can take place between different water parcels during their encounters. This exchange happens by diffusion and therefore relies on a concentration difference between two parcels. Thus, the intensity of mixing would depend on both the tracer distribution and the flow…*"

2. Lines 49–50: "Our method does not require the initial spacing between trajectories to be small ..." Mustn't the spacing always be sufficiently small to detect the relevant spatial features that determine the mixing properties? In other words, doesn't your method therefore require comparable spacings as other methods in order to properly capture the physics? Your examples in fact employ fairly dense spacings (see also remark below). Please comment.

*We agree with the reviewer and have removed the sentence in question from the revision.*

*Also, additional simulations have been carried out and included in the revised manuscript, which investigate the dependence of the encounter volume on the grid spacing. The following paragraph has been added to the revision at the end of Sec II(a):*

"We have carried out numerical simulations (Fig. 6) to investigate the dependence of the encounter volume on the grid size, and to come up with a rule of thumb recommendation regarding the appropriate grid spacing. Our simulations suggest that the encounter volume values (approximated by $V \approx N\,dV$) are relatively insensitive to the variations of grid spacing between 1/10 and 1/2 of the encounter radius (with the encounter radius being a fraction of the size of the feature of interest, as suggested by Fig. 2), and that the major effect of a coarser grid is the degraded resolution of the resulting V map, rather than incorrect V values."

[Figure]

**Figure 1. Encounter volume, V, for the Duffing Oscillator flow for various grids of initial positions, from dense grid spacing of 0.02 (left), to intermediate grid spacing of 0.04 (middle), to coarse grid spacing of 0.1 (right). Encounter radius, R=0.2, and integration time, T=6.67, are the same in all 3 simulations.**

*The emphasis on small spacing is also implied in the new text near the beginning of Sec I(b), where we formally define the encounter volume and number in the limit of infinitesimal particle spacing.*

3. Lines 62–67: encounter number $N$ is determined by the *first* encounter of a given trajectory with other trajectories. This relies on the assumption that in the absence of sources/sinks "most property exchange will occur during the first encounter" and in other cases "... the number of first encounters ... should still be relevant." This is a rather loose argumentation. May the concentration difference between parcel A and B (also in the absence of sources/sinks) not just as well be *larger* – causing *more* property exchange – on e.g. a *second* encounter due to property exchange of parcels and A and B with other parcels in between their first and second encounters? Please provide a stronger physical rationale for this first-encounter ansatz or present it more explicitly as an assumption or hypothesis.

*Following the reviewer's suggestion, in the revised manuscript we present this first-encounter ansatz as an assumption. The following sentence has been added to the revision to address this question:*

"Note that this assumption may not hold if the parcels re-acquire different properties after their first encounter due to encountering and exchanging properties with other parcels. In this case, or in the case when tracer variance is being continuously introduced, it may be more reasonable to count the total number of encounters."

4. Lines 84–86: "... encounter rates ... are locally the largest near a hyperbolic trajectory and along the segments of its associated stable manifolds ..." This exclusive link with stable manifolds is unclear. Don't the high encounter rates result from the rapid dispersion of fluid parcels due to exponential stretching in the homoclinic/heteroclinic tangles delineated by interacting (un)stable manifolds of hyperbolic points? In other words, don't stable and unstable manifolds contribute equally to the high encounter rates in chaotic regions? Hence, it seems more accurate to correlate regions of high $N$ with such tangles instead of only with stable manifolds. Please either better explain the (assumed) role of stable manifolds or link the behaviour with chaotic tangles.

*The reviewer might perhaps be referring to long integration times; in the long integration time limit, both stable and unstable manifolds densely fill the entire chaotic zone, and thus the entire chaotic zone is characterized by uniformly large encounter volume (equal to the volume of the*

*chaotic zone). So the reviewer is correct in saying that in the long integration time limit, there will be correlation between the entire manifold tangle and high encounter volume region.*

*Over short integration times, however, it is the stable manifold that acts as a pathway for bringing particles from remote regions into the vicinity of a hyperbolic trajectory, where particles stay over extended periods of time, and where many encounters occur. The unstable manifold, on the other hand, will rapidly remove a particle from the hyperbolic region, thus limiting its presence in the high-encounter region. Of course, the unstable manifolds will eventually bring a particle back into the vicinity of a hyperbolic region, but it will only do so intermittently (as the manifold will never reach the hyperbolic trajectory but instead will tangle around venturing away and coming back), and this process will require traveling along a significant portion of a homoclinic tangle (i.e., long integration time).*

*This exclusive link between forward-time integration and stable manifolds is not unique to encounter volume, but rather is typical for many finite-time methods, including FTLEs, which in forward time highlight stable manifold as maximizing ridges (see Fig. 2 below).*

[Figure]

**Figure 2. Comparison between the FTLEs (top) and the encounter volume (bottom; same as middle row of Fig. 2) for the Duffing Oscillator flow for various integration times, from T=0.1Tpert= 0.13 (on the left) to T=50Tpert=66.67 (on the right). The same set of trajectories, deployed on a dense initial grid with 0.02 grid spacing is used in all simulations. In the bottom panels, R=0.2.**

*To further clarify this issue, we have computed both stable and unstable manifolds for the Duffing Oscillator using a direct method, where we grew manifolds from a small segment starting at the hyperbolic trajectory (which in this example stays at the origin at all times.) Both manifolds were then superimposed on a forward-time encounter volume plot (see fig. 3 below). It is clear from this new simulation that in forward time high encounter volume correlates well with the stable (and not the unstable) manifolds.*

[Figure]

**Figure 3. Encounter Volume (color; the same as 2nd row and 2nd column subplot of Fig 2 in the paper) and stable (black) and unstable (white) manifolds for the Duffing Oscillator flow.**

*We have added the following 3 paragraphs to Sec I(b) and II(a) of the revised paper to address this question:*

"In the infinite time limit, $T \rightarrow \infty$, when all parcels within a chaotic zone (or turbulent region) of finite extent encounter all other parcels within the same chaotic zone, the encounter volume $V(\vec{x}_0, t_0; T \rightarrow \infty)$ approaches a constant equal to the volume (or area in 2d) of the chaotic zone. For 2D, incompressible flow, the encounter rates over finite $T$ are locally the largest near a hyperbolic trajectory and along the segments of its associated stable manifolds. The stable manifolds serve as pathways that bring water parcels from remote regions into the vicinity of the hyperbolic trajectory, where parcels stay for extended periods of time, and where many encounters occur. Note that the unstable manifolds, on the other hand, will rapidly remove a particle from a hyperbolic region, thus limiting its exposure to the high-encounter region near the hyperbolic trajectory. For this reason, the unstable manifolds are not revealed by encounter volume calculation performed in forward time and require a backward-time calculation instead. This exclusive link between forward/backward in time calculation of trajectories and stable/unstable manifolds, respectively, is not specific to the encounter volume diagnostic, but rather is typical for many finite-time methods from the dynamical systems theory, including finite-time Lyapunov exponents (FTLEs), which in forward time approximate segments of stable manifold as maximizing ridges (Haller, 2002; Shadden et al., 2005; Lekien and Ross, 2010)."

"In the long integration time limit, when each manifold, either stable or unstable, densely fills the entire chaotic zone forming a dense homoclininc or heteroclinic tangle, the whole tangle will be characterized by high encounter volumes in both forward and backward time. Again, this is similar to how the maximizing ridges of the forward time FTLEs elongate and sharpen with increasing integration time."

"In order to more clearly highlight the link between high values of $V$ and stable (rather than unstable) manifolds, we have computed both stable and unstable manifolds for the Duffing Oscillator flow using a direct method, where we grew manifolds from a small segment starting at the hyperbolic trajectory. For the Duffing Oscillator this computation is straightforward since the the hyperbolic trajectory stays at the origin at all times. Both stable and unstable directly-computed manifolds were then superimposed on a forward-time encounter volume plot in Fig. 4. The comparison shows that, as anticipated, the encounter volume diagnostic clearly highlights stable manifolds as maximizing ridges of $V$ computed in forward time."

5. Lines 93–94: "... will reveal longer segments of stable manifolds ... illustrated numerically in the next section." It actually more and more seems to reveal the abovementioned homoclinic/ heteroclinic tangles instead of the stable manifolds. Consider to this end the Duffing oscillator in Sec. IIa. Here the stable and unstable manifolds of the hyperbolic point form a pair symmetric about $x = 0$ (as remarked on line 126). Their interaction yields a homoclinic tangle that delineates a figure-8 region about the two islands in Fig. 1. This tangle – and thereby *both* manifolds – coincides with the region of highest encounter rates in Fig. 2. Results on the Bickley jet in Fig. 4 further seem to support this; here correlation actually occurs with the heteroclinic tangles delineated by the interacting (un)stable manifolds of the 3 hyperbolic points instead of only with the stable manifolds. Please comment and, if necessary, modify the discussion.

*Please see our answer to the previous comment and Figs. 2 and 3(Figs. 4 and 5 in the revised paper) above.*

6. The discussion of Fig. 2 implies that the encounter number indeed adequately captures the dynamics. However, to this end rather smooth distributions (as e.g. in Figs. 2–4) seem necessary, suggesting that the method requires a dense spacing of initial parcel positions in order to work properly. This contradicts the statement "... does not require the initial spacing between trajectories to be small ..." (lines 49-50). Moreover, this suggests that mixing analyses by the encounter number may in fact be far more expensive than standard Poincarje sectioning (typically requiring only a few dozen parcels). Please comment and, if necessary, modify the discussion.

*We have removed the sentence in question about the grid size and included numerical simulations that explore the dependence of encounter volume on the grid spacing (see our answer to comment 2 and Fig. 1 above).*

*We agree with the reviewer that the Poincare section is a powerful tool for revealing regular regions and chaotic zones in time-periodic flows. We also agree that it only requires a small number of parcels. However, its application is limited to time-periodic flows, and it requires*

*trajectories to be computed over very long integration times, typically thousands of periods of perturbation. The encounter volume, on the other hand, is not limited to time-periodic flows, and works with much shorter segments of trajectories (longest integration time in our simulations is only 50 periods of perturbation). It also is better suited for identifying the manifolds as it does not require any apriory knowledge about the location of the hyperbolic trajectory. The encounter volume, however, requires many more parcels to be released in order to map out the phase space than the Poincare section analysis. Thus, both methods have their own advantages and limitations. We have added this discussion to the revised paper.*

*At the end of Sec II(a) of the revised paper, we added a discussion of advantages and limitations of the Poincare section and encounter volume methods.*

"With a variety of dynamical systems techniques available, it is important to understand the advantages and limitation of the different methods. We compared the encounter volume to two well-established and commonly-used methods, the Poincare section (Fig. 3) and the FTLEs (Fig. 5). Since the Poincare section requires stroboscopic sampling of trajectories in time, it can only be applied to time-periodic flows, and requires that trajectories are computed over long integration time, typically thousands of the periods of the perturbation. On the other hand, it generally requires only a few parcels to be released at some key locations, rather than releasing a dense grid of initial positions, to map out the entire phase space. The encounter volume and FTLEs, on the other hand, are not limited to time-periodic flows, and also work with significantly shorter segments of trajectories (longest integration time in our simulations in Fig. 2 is only 50 periods of perturbation). They are also better suited for identifying manifolds than the Poincare sectioning as they do not require any *a priori* knowledge about the location of the hyperbolic trajectory. On the other hand, they require many more parcels to be released in order to map out the phase space. When applied to the same set of trajectories (same initial positions and integration times), the FTLEs and the encounter volume methods produced similar results (Fig. 5), with $V$ being arguably better suited for 1) identifying the coherent core regions of eddies, where FTLEs have spiraling patterns that complicate the analysis, and 2) producing more continuous segments of manifolds at intermediate integration times, when FTLE-based ridges get discontinuous near the turning points of a manifold. The advantage of FTLEs, on the other hand, is that they have fewer parameters ($T$ and grid spacing), whereas $V$ also depends on $R$, and that they less expensive computationally. The more expensive computational cost of $V$ compared to FTLEs is due to two reasons: first, the FTLEs only depend on the initial and final positions of trajectories, whereas $V$ depends on the entire trajectory history; and second, FTLEs depend on the relative distance between a trajectory and its closest neighbors, whereas $V$ keeps tracks of encounters with all trajectories, not just the neighboring trajectories. Thus, the cost of evaluating FTLE for each particle is independent of the total number of particles released, and the cost of evaluating $V$ for each particle increases in proportion to the number of particles (since one needs to keep track of encounters with all particles). The calculation of $V$ is still feasible for realistic geophysical flows, as is illustrated below. Note also that, depending on the physical question

being studied, the information about the entire trajectory, not just the final and initial position, might in fact be advantageous."

7. The above suggests that Poincarje sectioning outperforms the encounter-number method in periodic flows. Hence, the periodic examples mainly serve to demonstrate the physical validity of the encounter-number method; its true usefulness seems to be for essentially aperiodic flows as e.g. the Gulf Stream flow (Sec. IIc). However, the analysis of this flow is rather superficial and open-ended (lines 209–230). It is recommended to deepen this analysis so as to convincingly demonstrate the potential of the method (in particular) for aperiodic flows.

*Please see our answer to #6 regarding comparison and relative advantages/disadvantages of Poincare section vs encounter volume methods.*

*We agree with the reviewer that the Gulf Stream flow is the most interesting from the oceanographic point of view, and in the revised paper we have extended this section. However, in-depth analysis of transport and mixing properties of the Gulf Stream Extension flow, along with the study of the Lagrangian properties, leakiness and coherence of the Gulf Stream rings, is out of the scope of this paper, and will be better suited for a separate paper that will be devoted entirely to this topic.*

8. Sec. III: it is recommended to demonstrate validity of expressions (1), (2) and (9) for $N$ by comparison with $N$ found via actual parcel trajectories of the corresponding simplified flows.

*We have performed numerical simulations for the linear strain and linear shear flows to check the validity of formulas (2) and (9). Numerical simulations and analytical expressions for the encounter volume are in excellent agreement with each other (see Fig. 4 below). This figure was also added to the revised paper [Fig. 10 in the revised paper]).*

[Figure]

**Figure 4. Comparison between numerically computed encounter volume (blue) and analytical predictions (eqs. (8) and (9)) (red) for the linear strain (left) and linear shear flows (right). For the linear shear flow alpha=0.1, R=5, dx=dy=R/25; for the linear strain flow gamma=0.1, R=5; dx=dy=R/25.**

*The analysis of the diffusive flow appears to be more complicated than we anticipated. Our analytical expression (1) did not check out against numerical simulations and was removed from the paper. The difficulty is that our original treatment of the diffusive model as a process where the distance from the initial position for all trajectories (rather than on average) grows as square root of time was too simplistic, whereas other available probability laws and trends based on diffusive model do not account for the fact that we count only first encounters. Our simulations indicate growth of the encounter number in proportion to some power γ of time, where is in the range .64 to .78. But we are unable to predict the value of γ and have instead just quoted the range γ obtained. We have also described some of the initial steps that need to be taken towards resolution of the problem and we hope that some reader might be able to draw on this to make further progress. All of this appears in the first several paragraphs of Section III.*

9. Lines 310–311: "... vector flux of the scalar of interest. This linkage is made explicit by ..." This same concept of a net scalar flux (and corresponding trajectories) is in fact also adopted in studies on convective heat transfer and chemically-reacting flows [1, 2, 3, 4, 5]. Please mention this for a stronger connection with similar research and literature.

*We thank the reviewer for pointing out the references on the Lagrangian interpretation of thermal heat transfer. When we published our paper on the Lagrangian interpretation of scalar fluxes, we were unaware of this work. We have added the references as requested.*

10. Line 324: "Although this lack of uniqueness may seem troublesome ..." This ambiguity is in fact resolved in [5] by attaching physical validity to such an additional vector instead of treating it as an arbitrary field (see also remark below).

*See our response to #11*

11. Lines 347–348: "... it is most convenient to make use of the flexibility in the definition of the tracer flux ..." This suggests that the method produces arbitrary results and its physical meaning therefore is questionable. However, this approach can in fact be provided with a sound physical basis using the approach following [5]. Key to this is that, given linear transport equations, a scalar field $c$ governed by a transport equation of the form (10) admits expression as the difference between two other physically-meaningful scalar fields $c_A$ and $c_B$, each governed by $\partial c_A \partial t + \nabla \vee \mathbf{F}_A = S_A, \partial c_B \partial t + \nabla \vee \mathbf{F}_B = S_B$, (1)

with $\mathbf{F}_{A,B}$ and $S_{A,B}$ the corresponding fluxes and source terms, respectively. In [5], $S_A = S_B = 0$ and $\mathbf{F}_A$ and $\mathbf{F}_B$ are diffusive flux and advective-diffusive flux, respectively, of the same initial condition $c_A(\mathbf{x}, 0) = c_B(\mathbf{x}, 0) = g(\mathbf{x})$. In the current manuscript, also $S_A = S_B = 0$ yet $\mathbf{F}_A$ and $\mathbf{F}_A$ now both are the advective flux (i.e. $\mathbf{F}_{A,B} = \mathbf{u}c_{A,B}$) of the *different* initial conditions $c_A(\mathbf{x}, 0) = c_0$ and $c_B(\mathbf{x}, 0) = C_0(\mathbf{x})$. Hence, both problems, though physically different, allow for treatment by the same concept. Transport of difference $c = c_B - c_A$ is governed by $\partial c \partial t + \nabla \vee \mathbf{F}$

$= S', \mathbf{F} = \mathbf{F}_B - \mathbf{F}_A, S = S_B - S_A$, (2)

with here $S = 0$ and $\mathbf{F} = \mathbf{u}c = \mathbf{u}(c_B - c_A)$ the flux of $c$ (i.e. the anomaly and its flux in line

348). Thus anomaly $c$ in fact concerns the scalar transport relative to a physical reference state $c_A$ instead of some arbitrary state. Here the reference state happens to remain uniform in time due to the advective transport of a uniform initial condition, i.e. $c_A(\mathbf{x}, t) = c_A(\mathbf{x}, 0) = c_0$, yet the approach holds equally for any non-uniform (evolving) state $c_A$ (enabling its employment also for more complicated problems).[1] Moreover, note that $\mathbf{u}$ needn't be divergence-free. It is

recommended to modify the discussion in Sec. IV according to the above in order to eliminate the (incorrect) impression of a conceptual flaw in the method.

*We have recast our treatment of the dye flux in the Bickley jet as suggested, following the Speetjens (2012) arguments. We specifically introduce two scalar flux equations, one for c and one for the reference value $c_O$, and take the difference. This is indeed a more systematic approach, but we are not completely convinced that the flux vector so obtained by this methodology is always unique, or necessarily most desirable. The purely diffusive reference state that Speetjens favors may not be the only reference state that a reasonable person might choose: there could be alternatives. And in our problem $c_o$ is just a small (but arbitrary) value.*

**Minor technical issues and corrections**

1. Figs. 2–4: specify the spacing of the initial parcel positions.
2. Line 153: pronounces *!* pronounced
3. Line 231: the title of Sec. III is rather long and confusing. Please consider a more compact title.
4. Line 266: reference moving *!* reference frame moving

[1] A. Bejan, Convection Heat Transfer, Wiley, New York (1995).
[2] V.A.F. Costa, Bejan's heatlines and masslines for convection visualization and analysis, Appl. Mech. Rev. 59 (2006), 127.
[3] S. Mahmud, R. A. Fraser, Visualizing energy flows through energy streamlines and pathlines, Int. J. Heat Mass Transfer 50 (2007), 3990.
[4] A. Mukhopadhyay, X. Qin, S. K. Aggarwal, I. K. Puri, On extension of "heatline" and "massline" concepts to reacting flows through use of conserved scalars, ASME J. Heat Transfer 124 (2002), 791.
[5] M. F. M. Speetjens, A generalised Lagrangian formalism for thermal analysis of laminar convective heat transfer, Int. J. Therm. Sci. 61 (2012), 79.

[1]Reference state $\bar{T}$ in [5] e.g. corresponds with the non-uniform and unsteady temperature field due to diffusive heat transfer only; $\tilde{T} = T - \bar{T}$ is the contribution to the total advective-diffusive temperature field $T$ due to the flow $u$ (i.e. $\tilde{T} \neq 0$ only if $u \neq 0$) and thus captures the thermal transport that is effectively induced by the fluid motion.

*All the minor technical corrections have been made, and all new references added to the revised paper.*

In this paper the authors introduce a new Lagrangian descriptor to give a measure of the effectiveness of a flow to mix over a finite time. The idea is to start with a finite grid of K initial trajectories, and, for each trajectory compute the number other trajectories that come within a radius R of the given one, thus they compute

$$N(x, R, T, K) = \sum^{K} {k=1} I(\ \min_{s,t \in [0,T]} (\| \frown_s(x_k) - \frown_t(x)\|)\ \vee\ R)$$

where we define the indicator function $I$ to return 1 if true and 0 if false, and the flow $\frown_t(y) = x(t; y)$ for an initial point y. (The authors never give such a formula, and ignore the dependence on the gird).

While this is an intriguing idea, it is not clear how to make it mathematically well-defined. It seems to have some relation to finite time entropy, as introduced in the reference by Froyland as this computes the growth rate of number of distinguishable trajectories. Would it be better to talk about a growth-rate here too? I feel that one should not just compute something that is so specific to choices, but first make a consistent mathematical definition: something that exists in the limit as the grid of initial points becomes infinitely fine, say, and then compute it, showing that the computations are, to some approximation, giving the desired quantity.

*Following the reviewer's suggestion, we reformulated the mixing potential concept in terms of the trajectory encounter mass, M, and its simplified approximation – the encounter volume, V, which we now define in a continuous limit of infinitely many infinitesimally small water parcels or, equivalently, an infinitely dense grid of initial positions. For an incompressible flow densely seeded with particles, the encounter volume V can be approximated by $V \cong N\delta V$, where N is the encounter number, i.e., the number of trajectories passing through an encounter sphere of radius R moving with the parcel over time T, and $\delta V$ is a parcel volume element. We have also included a mathematical expression for the encounter number as the reviewer suggested above. The beginning of Sec I(b) of the revised paper now reads:*

"For a given reference trajectory, $\vec{x}(\vec{x}_0, t_0; T)$, the *encounter mass, $M(\vec{x}_0, t_0; T)$,* is defined as the total mass of fluid that passes within a radius $R$ of reference trajectory over a finite time interval $to < t < to + T$. One might imagine a sphere that has radius $R$ and that is centered at and moves with the reference trajectory. The encounter mass then consists of the mass of the fluid that is initially located within the sphere along with the mass of all the fluid that passes through the sphere over the time interval $to < t < to + T$. Note that it is generally not possible to compute the latter by simply integrating the mass flux into the sphere over $to < t < to + T$ since some fluid may leave and then re-enter the sphere and would be counted more than once, so Lagrangian information is required to keep track of the history of each fluid parcel trajectory entering the sphere.

To this end, subdivide the entire fluid at $t = to$ into small compact fluid elements with masses $\delta M_i = \rho_i \delta V_i$, where $\rho_i$ is the density of a fluid element and $\delta V_i$ is its volume. We wish to follow the motion of each fluid element over time interval $to < t < to + T$, and we assume that the

elements remain compact over such time, so that the motion of each fluid element can be well-represented by one trajectory. If the fluid elements stretch and deform too much, we can evoke the continuum hypothesis and make $\delta M$ sufficiently small that such compactness is assured. In the limit of infinitesimal fluid elements, $\delta M_i \rightarrow dM$, we can associate with each infinitesimal fluid element a unique trajectory. The encounter mass is then

$$M = \lim_{dM_i \rightarrow 0} \Sigma_i \, dM_i.$$

For an incompressible flow, the density and volume of each fluid element, $\rho_i$ and $\delta V_i$, remain constant following a trajectory, although different fluid elements are still allowed to have different densities such as, for example, in stratified 3D geophysical flows. If the flow is unstratified, the densities of all fluid elements are equal, $\rho_i = \rho$, and the encounter mass becomes

$$M = \rho \, V,$$

where

$$V(\vec{x}_0, t_0; T) = \lim_{dV_i \rightarrow 0} \Sigma_i \, dV_i$$

is the *encounter volume* – the total volume of fluid that passes within a radius $R$ of reference trajectory over a finite time interval $to < t < to + T$. When all volume elements are equal, $dV_i = dV$, the encounter volume can be further simplified to

$$V = \lim_{dV \rightarrow 0} N dV,$$

where the *encounter number*, $N(\vec{x}_0, t_0; T)$, is the number of trajectories that come within a radius $R$ of the reference trajectory over a time interval $to < t < to + T$. We will refer to $t_0$ as the starting time, $T$ as the trajectory integration time, and $\vec{x}_0$ as the trajectory initial position, i.e., $\vec{x}(\vec{x}_0, t_0; T = 0) = \vec{x}_0$. For practical applications with geophysical flows, the limit in the definition of the encounter volume can be dropped and one can simply use the approximation

$$V \approx N \, \delta V$$

with the dense grid of initial positions $\vec{x}_0$. Mathematically, the encounter number can be written as $N = \sum_{k=1}^{K} I(\min(|\vec{x_k}(\vec{x}_0, t_0; T) - \vec{x}(\vec{x}_0, t_0; T)|) \leq R)$ where the indicator function I is 1 if true and 0 if false, and K is the total number of Lagrangian particles released. The encounter volume depends on the starting time, integration time, encounter radius, and the number of trajectories (i.e., grid spacing); all of these parameter dependences will be discussed below. Once the encounter volume is estimated, regions of space with large/small $V$ would then be associated with enhanced/inhibited mixing potential."

1. The authors fix the grid size and do not investigate how the number depends on grid size. The do not even tell the reader what grids are used in the first two examples!

*We apologize for not providing the grid size that we used in the first two examples. This info has been included in the revision.*

*In the revised paper, we also present a set of new numerical simulations exploring the dependence of the encounter volume on grid size. The following paragraph discussing this issue has been added to the revision at the end of Sec II(a):*

"We have carried out numerical simulations (Fig. 6 [Fig. 5 here]) to investigate the dependence of the encounter volume on the grid size, and to come up with a rule of thumb recommendation regarding the appropriate grid spacing. Our simulations suggest that the encounter volume values (approximated by $V \approx N \delta V$) are relatively insensitive to the variations of grid spacing between 1/10 and 1/2 of the encounter radius (with the encounter radius being a fraction of the size of the feature of interest, as suggested by Fig. 2), and that the major effect of a coarser grid is the degraded resolution of the resulting $V$ map, rather than incorrect $V$ values. "

[Figure]

**Figure 5. Encounter volume, V, for the Duffing Oscillator flow for various grids of initial positions, from dense grid spacing of 0.02 (left), to intermediate grid spacing of 0.04 (middle), to coarse grid spacing of 0.1 (right). Encounter radius, R=0.2, and integration time, T=6.67, are the same in all 3 simulations.**

2. It seems like it would be better to define something that (like they mention in the conclusions) represents a "fraction" or "density" of encounters. Mathematically one would probably define something that uses an ⌢ − ⌢ construction: Given trajectories on an ⌢ grid, how many get closer than ⌢? Then take limits. if possible, of a density or growth-rate?

*As explained above, we now quantify mixing potential using the encounter volume, V, instead of the encounter number. The encounter radius, which defines how close to each other two parcels need to be in order to be counted as an encounter, is kept finite and treated as a parameter. The dependence of V on R is investigated numerically, and analytical arguments are presented that relate V, R and grid spacing to the size of the features of interest, all in agreement with numerical simulations.*

3. Another possible quantity, though instead of measuring "mixing" would be one that measures "ergodicity": How many grid cells does a given trajectory cover? This might also be an interesting quantity, and much easier to compute. Note that mixing is equivalent to each trajectory visiting every grid cell.

*The relationship between Lagrangian Coherent Structures and ergodicity has been explored in our prior work, please see the following paper for the discussion of this topic: Rypina, I. I., S.*

*Scott, L. J. Pratt, and M. G. Brown (2011). Investigating the connection between complexity of isolated trajectories and Lagrangian coherent structures. Nonlin. Proc. Geophys., 18, 977-987, doi:10.5194. This reference has been added to the revised paper.*

4. The authors do not really compare their results with any of the other many possible descriptors like FTLE, or perhaps more relevantly the finite time entropy.

*At the end of Sec II(a) of the revised paper, we added a comparison between encounter volume V and FTLEs, along with a discussion of the advantages and limitations of both methods.*

"With a variety of dynamical systems techniques available, it is important to understand the advantages and limitation of the different methods. We compared the encounter volume to two well-established and commonly-used methods, the Poincare section (Fig. 3) and the FTLEs (Fig. 5 [Fig. 6 here]). Since the Poincare section requires stroboscopic sampling of trajectories in time, it can only be applied to time-periodic flows, and requires that trajectories are computed over long integration time, typically thousands of the periods of the perturbation. On the other hand, it generally requires only a few parcels to be released at some key locations, rather than releasing a dense grid of initial positions, to map out the entire phase space. The encounter volume and FTLEs, on the other hand, are not limited to time-periodic flows, and also work with significantly shorter segments of trajectories (longest integration time in our simulations in Fig. 2 is only 50 periods of perturbation). They are also better suited for identifying manifolds than the Poincare sectioning as they do not require any *a priori* knowledge about the location of the hyperbolic trajectory. On the other hand, they require many more parcels to be released in order to map out the phase space. When applied to the same set of trajectories (same initial positions and integration times), the FTLEs and the encounter volume methods produced similar results (Fig. 5), with $V$ being arguably better suited for 1) identifying the coherent core regions of eddies, where FTLEs have spiraling patterns that complicate the analysis, and 2) producing more continuous segments of manifolds at intermediate integration times, when FTLE-based ridges get discontinuous near the turning points of a manifold. The advantage of FTLEs, on the other hand, is that they have fewer parameters ($T$ and grid spacing), whereas $V$ also depends on $R$, and that they less expensive computationally. The more expensive computational cost of $V$ compared to FTLEs is due to two reasons: first, the FTLEs only depend on the initial and final positions of trajectories, whereas $V$ depends on the entire trajectory history; and second, FTLEs depend on the relative distance between a trajectory and its closest neighbors, whereas $V$ keeps tracks of encounters with all trajectories, not just the neighboring trajectories. Thus, the cost of evaluating FTLE for each particle is independent of the total number of particles released, and the cost of evaluating $V$ for each particle increases in proportion to the number of particles (since one needs to keep track of encounters with all particles). The calculation of $V$ is still feasible for realistic geophysical flows, as is illustrated below. Note also that, depending on the physical question being studied, the information about the entire trajectory, not just the final and initial position, might in fact be advantageous."

[Figure]

**Figure 6. Comparison between the FTLEs (top) and the encounter volume (bottom; same as middle row of Fig. 2) for the Duffing Oscillator flow for various integration times, from T=0.1Tpert= 0.13 (on the left) to T=50Tpert=66.67 (on the right). The same set of trajectories, deployed on a dense initial grid with 0.02 grid spacing is used in all simulations. In the bottom panels, R=0.2.**

5. The authors do not discuss the complexity of this computation. It seems to me that it is much more computationally intensive than, e.g., the FTLE, which does not involve comparing all distances between all trajectories. It this really a feasible calculation? How does it scale with the number of trajectories and the time?

*We have added a discussion of the complexity of the calculation and scaling with the number of trajectories. The reviewer is certainly correct that the encounter volume diagnostic is more computationally expensive than FTLEs. However, the calculation of the encounter volume is certainly feasible for realistic oceanic flows, as illustrated in our data-based example #3 (satellite-based geostrophic velocities). Please see our answer to comment 4 above.*

6. The authors do some basic investigation of how $N$ depends upon $R$ and $t$, but the computations of $N$ for the simple diffusive and shear cases seem wrong to me: In particular, if we take a planar diffusive process with diffusion coefficient $D$, and make the assumption (not clear to me) that one can transform to a frame moving with one particle (doesn't this double the diffusivity?), …

*We thank the reviewer for pointing out that the diffusivity is doubled in the reference frame moving with a particle. This has been corrected in the revised paper.*

…then one should compute the probability of a particle finding itself inside a disk of radius R for any time $0 < t < T$, given it starts at some point $(x_0, y_0)$ in the plane. For example for the process on the line, then at a FIXED time $t$ this means evaluating the integral

$$P(\clubsuit x(t) \clubsuit < R \clubsuit x(0) = x_0) = \frac{1}{2\sqrt{Dt}} \int_{-R}^{R} \exp\sqrt{-\frac{(x - x_0)^2}{2Dt}} dx$$

which can be evaluated in terms of error functions. The authors seem to assume a deterministic motion with the root mean square distance, which seems to me to be wrong.

*We did not assume a deterministic motion, but our treatment of the diffusive motion was indeed incorrect and our analytical expression (1) disagreed with numerical simulations. We have removed expression (1) from the revised paper. Also see our response to comment #8 of the first reviewer.*

They also ignore particles that start inside the circle of radius $R$ (not so important if they want a large $t$ limit I suppose).

*The reviewer is correct; we did not include the volume of the encounter sphere (or the area of the encounter circle in our 2d examples) in our encounter volume calculations/formulas. We have clarified this in the revised paper, and noted that "To include the volume of fluid that is initially located within the encounter sphere (or within the encounter circle in this 2D case), one needs to add $\pi R^2$ to expression (2). The contribution of this term gets negligibly small as $T -> \infty$."*

Now to compute $N$ you have to sum (or integrate?) this probability over an initial distribution of initial points, say $x_0$ is uniform on a box, perhaps? And you have to somehow compute the probability over all times $0 < t < T$. This calculation seems very different from the one given in the paper.

*The formula for the pdf of a particle position that the reviewer wrote above does not take into account that we are interested in first encounters, not all encounters. In the revised paper, we have outlined some initial steps towards deriving the connection between V and diffusivity along the lines suggested by the reviewer, but we have not been able to follow through with this derivation; this is left for a future study.*

The shear flow is easier, but I think not done correctly either. One has to compute the area of the region that sweeps into the circle of radius R, but also include the particles that start inside the disk.

*Again, the reviewer is correct; our expression did not include the area of the encounter sphere, $\pi R^2$. We have added a note on this similar to the one for the strain flow.*

While this paper has an intriguing idea, I think it needs substantial revision and correction before publication.

*We are glad that the reviewer found our ideas intriguing, and we hope that we addressed all of the reviewer's concerns in the revised paper.*

It is only the short comment, not deep review. We can see many finite time Lagrangian descriptors suggested last time, maybe 20 or 30 years. For example, the Poincare section is calculated for finite time. Most descriptors follow from dynamical systems theory constructions. It well discussed in reviewers comments. I have an analogous question. Is it possible to make some connection between the encounter number and the Poincare recurrence. In any case, I think, the most Lagrangian descriptors used, for example, in oceanography have no strong mathematical foundation, but very useful for data analyze and interpretation. I also think the descriptor suggested here is more expensive in comparison with FLTE or some other. But it seems, it has some advantage in physical interpretation. I think the manuscript is suitable for publication in Nonlinear Processes in Geophysics, after some revision.

*A discussion of the differences and similarities between the encounter volume and the Poincare section methods have been added to the revised manuscript. We also added a discussion of the computational cost of the calculation, and a comparison with FTLEs. The new text at the end of Sec. II(a) now reads:*

"With a variety of dynamical systems techniques available, it is important to understand the advantages and limitation of the different methods. We compared the encounter volume to two well-established and commonly-used methods, the Poincare section (Fig. 3) and the FTLEs (Fig. 5). Since the Poincare section requires stroboscopic sampling of trajectories in time, it can only be applied to time-periodic flows, and requires that trajectories are computed over long integration time, typically thousands of the periods of the perturbation. On the other hand, it generally requires only a few parcels to be released at some key locations, rather than releasing a dense grid of initial positions, to map out the entire phase space. The encounter volume and FTLEs, on the other hand, are not limited to time-periodic flows, and also work with significantly shorter segments of trajectories (longest integration time in our simulations in Fig. 2 is only 50 periods of perturbation). They are also better suited for identifying manifolds than the Poincare sectioning as they do not require any *a priori* knowledge about the location of the hyperbolic trajectory. On the other hand, they require many more parcels to be released in order to map out the phase space. When applied to the same set of trajectories (same initial positions and integration times), the FTLEs and the encounter volume methods produced similar results (Fig. 5), with $V$ being arguably better suited for 1) identifying the coherent core regions of eddies, where FTLEs have spiraling patterns that complicate the analysis, and 2) producing more continuous segments of manifolds at intermediate integration times, when FTLE-based ridges get discontinuous near the turning points of a manifold. The advantage of FTLEs, on the other hand, is that they have fewer parameters ($T$ and grid spacing), whereas $V$ also depends on $R$, and that

they less expensive computationally. The more expensive computational cost of $V$ compared to FTLEs is due to two reasons: first, the FTLEs only depend on the initial and final positions of trajectories, whereas $V$ depends on the entire trajectory history; and second, FTLEs depend on the relative distance between a trajectory and its closest neighbors, whereas $V$ keeps tracks of encounters with all trajectories, not just the neighboring trajectories. Thus, the cost of evaluating FTLE for each particle is independent of the total number of particles released, and the cost of evaluating $V$ for each particle increases in proportion to the number of particles (since one needs to keep track of encounters with all particles). The calculation of $V$ is still feasible for realistic geophysical flows, as is illustrated below. Note also that, depending on the physical question being studied, the information about the entire trajectory, not just the final and initial position, might in fact be advantageous."

I can find few misprints in the manuscript which not mentioned by reviewers.
*We have carefully checked the revised manuscript for the misprints and typos.*

Minor technical issues and correction lines 336,371-373 t0 is used and may be better to use $t_0$
Caption to figure 8: U-star and Nstar are used. Maybe it will be better to use $U_\vee$ and $N_\vee$

*We have changed t0 to $t_0$, but changed Ustar and Nstar to $u^*$ and $N.^*$*